# Steering surface reconstruction of copper with electrolyte additives for $CO_2$ electroreduction

Zishan Han[1,2,6], Daliang Han [1,2,6], Zhe Chen [3,6], Jiachen Gao[1,2], Guangyi Jiang[1,2], Xinyu Wang[1,2], Shuaishuai Lyu[1,2], Yong Guo[1,2], Chuannan Geng[1,2], Lichang Yin [3,4✉], Zhe Weng [1,2✉] & Quan-Hong Yang [1,2,5✉]

Electrocatalytic $CO_2$ reduction to value-added hydrocarbon products using metallic copper (Cu) catalysts is a potentially sustainable approach to facilitate carbon neutrality. However, Cu metal suffers from unavoidable and uncontrollable surface reconstruction during electrocatalysis, which can have either adverse or beneficial effects on its electrocatalytic performance. In a break from the current catalyst design path, we propose a strategy guiding the reconstruction process in a favorable direction to improve the performance. Typically, the controlled surface reconstruction is facilely realized using an electrolyte additive, ethylenediamine tetramethylenephosphonic acid, to substantially promote $CO_2$ electroreduction to $CH_4$ for commercial polycrystalline Cu. As a result, a stable $CH_4$ Faradaic efficiency of 64% with a partial current density of 192 mA cm$^{-2}$, thus enabling an impressive $CO_2$-to-$CH_4$ conversion rate of 0.25 μmol cm$^{-2}$ s$^{-1}$, is achieved in an alkaline flow cell. We believe our study will promote the exploration of electrochemical reconstruction and provide a promising route for the discovery of high-performance electrocatalysts.

[1] Nanoyang Group, State Key Laboratory of Chemical Engineering, School of Chemical Engineering and Technology, Tianjin University, Tianjin 300072, China. [2] Haihe Laboratory of Sustainable Chemical Transformations, Tianjin 300192, China. [3] Shenyang National Laboratory for Materials Science, Institute of Metal Research, Chinese Academy of Sciences, 72 Wenhua Road, Shenyang 110016, China. [4] Department of Physics and Electronic Information, Huaibei Normal University, Anhui, Huaibei 235000, China. [5] Joint School of National University of Singapore and Tianjin University, International Campus of Tianjin University, Binhai New City, Fuzhou 350207, China. [6] These authors contributed equally: Zishan Han, Daliang Han, Zhe Chen. ✉email: lcyin@imr.ac.cn; zweng@tju.edu.cn; qhyangcn@tju.edu.cn

The continuously increasing carbon dioxide ($CO_2$) emissions caused by huge fossil fuel consumption lead to severe environmental changes associated with global warming. The $CO_2$ electroreduction reaction ($CO_2RR$) is attractive as a sustainable effort to recycle $CO_2$ into valuable industrial feedstocks and fuels for reducing the greenhouse effect[1]. Among all the catalysts for the $CO_2RR$, copper (Cu) metal is uniquely capable of producing multi-electron transfer hydrocarbon products with appreciable activity[2,3], however, its catalytic activity and selectivity for a certain product are still far from satisfactory[4,5]. Because of their weak cohesive energy and high surface mobility, metallic Cu catalysts usually undergo reconstruction involving atomic re-arrangement and compositional change during electrocatalysis[6–9]. Since the electrocatalytic properties have a noticeable dependence on the surface structure, the reconstruction behavior of Cu metal greatly impacts its $CO_2$ electroreduction performance[10–12].

The reconstruction behavior of Cu is a double-edged sword for the $CO_2RR$. Several studies have shown that the initial well-defined morphology and highly active sites tend to be lost during electrocatalysis because of surface reconstruction, resulting in catalytic performance degradation[13,14]. Nevertheless, surface reconstruction sometimes generates unique active sites with increased catalytic activity for specific products. Our previous work has shown that the reconstruction of a Cu complex under working conditions causes the formation of Cu nanoclusters with high selectivity for $CH_4$[9]. Kim et al. found that densely arranged Cu nanoparticles undergo reconstruction to in-situ form a disordered Cu nanostructure with a high selectivity for multicarbons[15]. Unfortunately, previous findings indicate that catalytic activity induced by restructuring is hardly accessible through ex-situ paths. Even if it could be, the well-designed catalysts are still at the risk of oxidation or poisoning in air, or deactivation induced by the restructuring process[16,17].

Steering the dynamic reconstruction process in a favorable direction may be a strategy for improving catalytic performance. However, due to the lack of mechanistic understanding, there is currently almost no effective way to control the reconstruction of Cu. Recently, Li et al. showed that the surface atomic migration is driven by the interplay of electric fields and adsorbed reaction intermediates for cathodic metal electrocatalysts[18]. Except for the reaction intermediates, we have shown that surface adsorbed/decorated species can also be introduced artificially to facilitate surface reconstruction[12]. Selective surface-capping additives have been widely used to guide the preferential exposure of crystal faces of the products in chemical synthesis[19,20], it is thus of great potential as a simple but effective way to control surface reconstruction of metallic Cu catalysts for a practical $CO_2RR$ technology.

Here, we show our effort on the electrolyte additive way to well guide the surface reconstruction of commercial polycrystalline Cu (poly-Cu), and the additive presented here is ethylenediamine tetramethylenephosphonic acid (EDTMPA) to help achieve an excellent $CO_2$ electroreduction performance. Experimental and theoretical studies show that EDTMPA molecules are preferentially adsorbed on Cu(110) during the $CO_2RR$, which not only induces the selective generation of Cu(110) facets that have an inherently high *CO binding strength, but also forms a local environment that promotes proton transfer from water to the Cu(110) facets. Since the rate-determining step of $CH_4$ formation is *CO + *H → *CHO[21], the simultaneously increased *CO and proton supplies together with *CHO stabilization caused by EDTMPA result in an excellent electrocatalytic $CO_2$ reduction performance for $CH_4$ production. In an H-cell configuration, commercial poly-Cu exhibits a $CO_2$-to-$CH_4$ Faradaic efficiency (FE) of 61% with a partial current density of 25 mA cm$^{-2}$ at −1.0 V versus a reversible hydrogen electrode (RHE) in a 0.5 M potassium bicarbonate ($KHCO_3$) electrolyte with 8 ppm EDTMPA, greatly outperforming the EDTMPA-free case. Even in an alkaline flow-cell configuration, where the $CO_2$-to-$CH_4$ selectivity is usually completely suppressed according to previous reports[22,23], our strategy is still highly efficient in obtaining a nearly constant $CH_4$ FE of 64 ± 2% at an operating current density of 300 mA cm$^{-2}$ over 6 h. Consequently, a $CO_2$-to-$CH_4$ conversion rate of 0.25 μmol cm$^{-2}$ s$^{-1}$ is achieved for commercial Cu-based electrocatalysts.

## Results

**$CO_2$ electroreduction performance.** Electrocatalysis was performed at −1.0 V versus RHE in $CO_2$-saturated 0.5 M $KHCO_3$ electrolytes with/without EDTMPA using an H-cell configuration (Fig. 1 and Supplementary Fig. 1). In the electrolyte with a trace

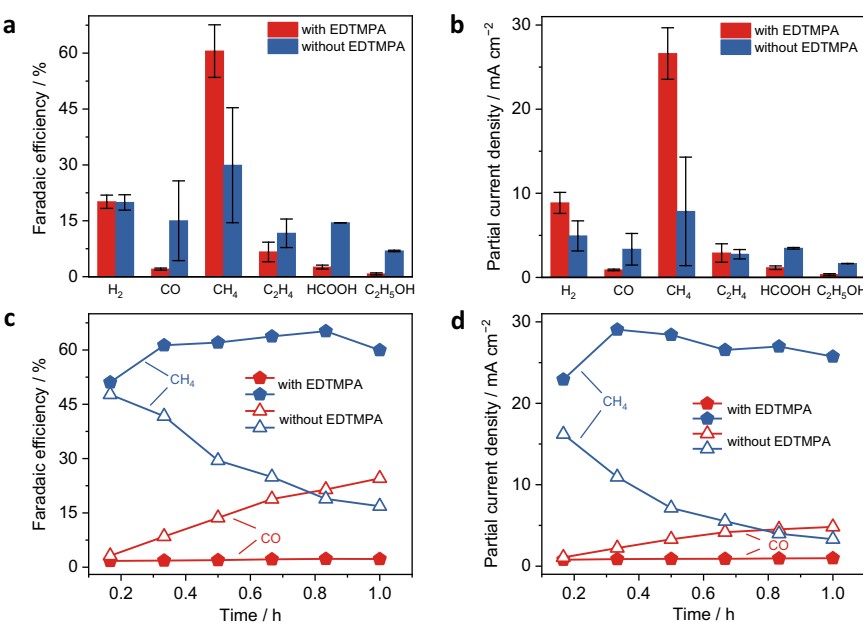

**Fig. 1 $CO_2$ electroreduction performance in an H-cell. a, b** Comparison of average FEs (**a**) and partial current densities (**b**) of various products at −1.0 V versus RHE in the electrolytes with and without EDTMPA during one-hour testes. The error bars in **a** and **b** represent mean absolute deviation. **c, d** Stability of FEs (**c**) and partial current densities (**d**) of the CO and $CH_4$ products in the electrolytes with and without EDTMPA.

amount (8 ppm) of EDTMPA, a commercial poly-Cu electrode has a much higher average $CH_4$ FE and partial current density (61% and 26.6 mA cm$^{-2}$) than those (30% and 7.8 mA cm$^{-2}$) in the EDTMPA-free electrolyte (Fig. 1a, b) during one-hour tests. EDTMPA also greatly increases the catalytic stability of poly-Cu electrodes during continuous electrocatalysis. In the EDTMPA-free case, the poly-Cu electrode retained only 35% and 20% of the initial FE and partial current density of $CH_4$, respectively, accompanied by gradually increased CO production (Fig. 1c, d). In sharp contrast, with the help of EDTMPA, the poly-Cu electrode showed an impressively stable FE and partial current density of $CH_4$ and a negligible CO increase (Fig. 1c, d). Considering that surface roughness can affect the electrocatalytic selectivity and activity[24], the roughness factors of the poly-Cu electrodes were estimated from the cyclic voltammetry (CV) measurements. As a result, both the poly-Cu electrodes become rough and have similar roughness factors of 1.61 and 1.25 after electrocatalysis in the electrolytes with and without EDTMPA, respectively (Supplementary Fig. 2), which eliminates the influence of surface roughness on the increased $CO_2$ electroreduction performance for $CH_4$ production.

**Generation of Cu(110) induced by EDTMPA.** Scanning electron microscope (SEM) images confirm that the originally smooth surface of the commercial poly-Cu electrode became rough after electrocatalysis in both electrolytes with and without EDTMPA, however, the resulting surface morphologies are significantly different (Fig. 2a and Supplementary Fig. 3). For the one tested in the EDTMPA-free electrolyte, its rough surface has some aggregated irregular nanoparticles (Supplementary Fig. 3c, d). In sharp contrast, massive uniformly distributed polyhedral nanoparticles were observed for the one tested in the EDTMPA-added electrolyte (Fig. 2a and Supplementary Fig. 3e, f), which is different from the

resulting cubic nanoparticles enclosed by {100} facets in previous studies[13,25,26].

Since the commercial poly-Cu electrodes are difficult to prepare for the transmission electron microscope (TEM) characterization, we used electrodeposited Cu TEM grids as poly-Cu electrodes for the $CO_2$RR and probed their crystal structure before and after electrocatalysis. Representative TEM images of the Cu grid after electrocatalysis in the EDTMPA-added electrolyte show a lot of Cu nanocrystals with hexagonal and cubic outlines, in good agreement with the ideal projections of a rhombic dodecahedral model bounded by {110} facets from different directions[27] (Fig. 2b and Supplementary Fig. 4). The Cu rhombic dodecahedrons are further confirmed by the high-resolution TEM (HRTEM) image and the corresponding selected area electron diffraction (SAED) pattern of the equilateral hexagonal projection shape of Cu nanocrystals along [111][27] (Fig. 2c). However, irregular Cu nanoparticles without any preferential surface orientation are merely observed for the electrodeposited Cu TEM grids before and after electrocatalysis in the EDTMPA-free electrolyte (Supplementary Figs. 5, 6).

Grazing incidence X-ray diffraction (GIXRD) patterns show no surface phase other than Cu, and that the intensity of the Cu(220) peak at 74.1º (JCPDS No. 04-0836) increases significantly for the poly-Cu electrode tested in the EDTMPA-added electrolyte (Fig. 2d). Its (220) to (111) peak intensity ratio is 4× that of the one tested in the EDTMPA-free electrolyte, which indicates that more Cu(110) surface planes are formed during electrocatalysis in the EDTMPA-added electrolyte.

The surface structure of the poly-Cu electrodes was further probed using the hydroxide (OH$^-$) electroadsorption technique[28,29]. Compared to the poly-Cu electrode tested in the EDTMPA-free electrolyte, the one tested in the EDTMPA-added electrolyte shows a pronounced Cu(110) OH$^-$ adsorption peak at ~0.4 V versus RHE in its linear sweep voltammetry profile (Fig. 2e), suggesting a high

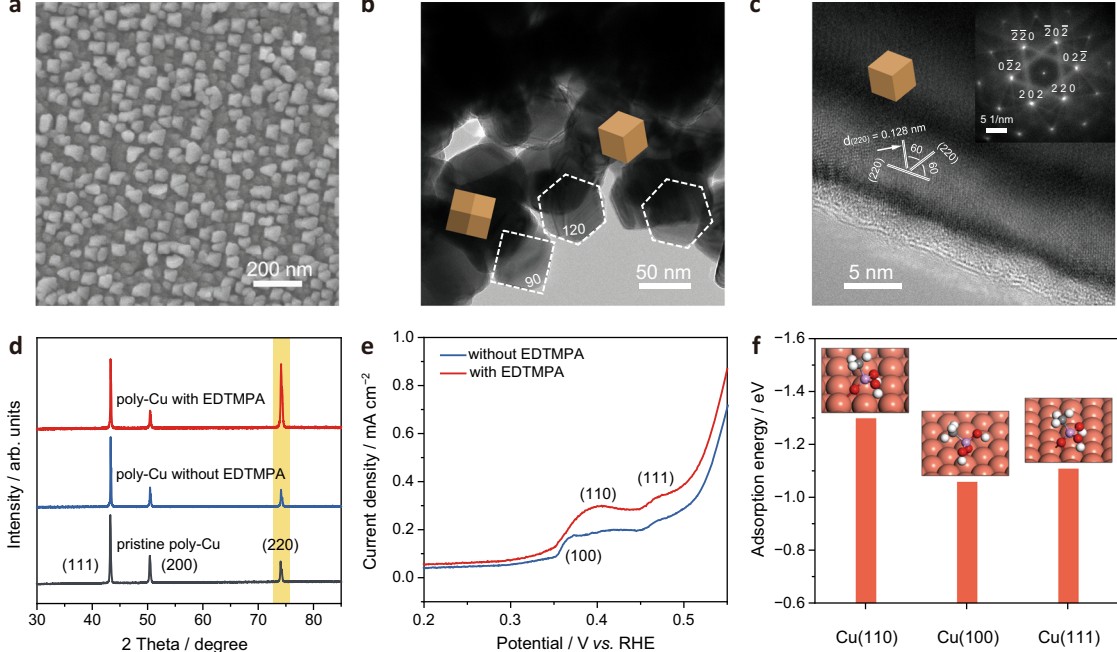

**Fig. 2 Characterization of the poly-Cu electrodes in different electrolytes. a** SEM image of a poly-Cu electrode after electrocatalysis in the EDTMPA-added electrolyte. **b, c** TEM image (**b**) and HRTEM image (**c**) of an electrodeposited Cu TEM grid after electrocatalysis in the EDTMPA-added electrolyte. Inset in **c** shows the corresponding SAED pattern. **d** GIXRD patterns of poly-Cu electrodes before/after electrocatalysis in electrolytes with and without EDTMPA. **e** OH$^-$ electroadsorption profiles on the poly-Cu electrodes after electrocatalysis in electrolytes with and without EDTMPA at a sweep rate of 100 mV s$^{-1}$ in 1 M KOH. **f** DFT-calculated adsorption energies of an MPA molecule on Cu(110), Cu(100) and Cu(111) surfaces. Insets in **f** are the corresponding atomic structure models.

surface density of (110). All the above results confirm that the EDTMPA induces the in-situ generation of Cu(110) surfaces during the reconstruction process.

Density functional theory (DFT) simulations were performed to understand the role of EDTMPA in the generation of Cu(110). To simplify the calculations, the $-CH_4O_3P$ fragment of an EDTMPA molecule ($C_6H_{20}N_2O_{12}P_4$) was taken with a H atom to passivate the terminal C atom as a methanephosphonic acid (MPA, $CH_5O_3P$) molecule, which was used to represent the huge EDTMPA molecule (Supplementary Fig. 7). The DFT results show that the MPA molecule has a higher adsorption energy on Cu(110) ($-1.30$ eV) than on Cu(100) ($-1.06$ eV) and Cu(111) ($-1.11$ eV) (Fig. 2f and Supplementary Figs. 8, 9), indicating that EDTMPA molecules prefer to be adsorbed on Cu(110). The adsorbed EDTMPA molecules stabilize the intrinsically high-energy Cu(110), and thus (110) surfaces are preferentially generated on the poly-Cu electrode during electrocatalysis in the EDTMPA-added electrolyte.

**Adsorption of EDTMPA molecules on Cu(110).** First of all, the stability of EDTMPA molecules is examined by CV tests, which were performed on poly-Cu electrodes between 0.3 V and $-1.7$ V versus RHE in Ar-saturated 0.5 M KHCO$_3$ electrolytes without and with a high concentration (100 ppm) of EDTMPA. The CV curves in both electrolytes are nearly identical, which confirms that the EDTMPA molecule is quite stable in the wide electrochemical window (Supplementary Fig. 10). The adsorption of EDTMPA was verified by X-ray photoelectron spectroscopy (XPS). Obvious N and P elements were detected for the poly-Cu electrode tested in the EDTMPA-added electrolyte, but no for the one soaked in the EDTMPA-added electrolyte for the same time at an open circuit potential (Supplementary Fig. 11). It is indicated that EDTMPA has a potential-driven specific adsorption behavior on the Cu surface[30], which was further confirmed by in-situ Raman measurement (Supplementary Fig. 12). The Cu−O vibration bands ($600 - 620$ cm$^{-1}$)[31] are present in the EDTMPA-added case, and the blue shift occurs as the cathodic polarization increases, indicating that the EDTMPA molecules are coordinated at the Cu surface via Cu−O bonds and the bonding strength is proportional to the cathodic polarization, the same as revealed by DFT calculations (Supplementary Figs. 7, 8 and Supplementary Table 1).

The coverage of EDTMPA molecules on Cu(110) was also investigated to rule out the possibility of blocking the active sites for the CO$_2$RR. A complete EDTMPA molecule was structurally relaxed on Cu(110) surface with a 2×3 supercell, which represents the close-packed structure of EDTMPA molecules on Cu(110) (Supplementary Fig. 13a, b). Obviously, an adsorbed EDTMPA molecule only occupies up to two Cu atoms per six ones, while other Cu atoms are available for the CO$_2$RR (marked with green circles in Supplementary Fig. 13a, b). Furthermore, we explored the transfer process of a CO$_2$ molecule through the EDTMPA adsorption layer on Cu(110) (Supplementary Fig. 13c, d). DFT results show that it is an exothermic process, indicating that CO$_2$ molecules can approach the Cu(110) surface without obstacle. Therefore, the adsorbed EDTMPA molecules do not block the CO$_2$RR.

**High *CO coverage on Cu(110).** Previous studies argued that Cu(111) rather than Cu(110) favors the CH$_4$ yield during the CO$_2$RR[32,33], whereas Cu(110) seems to promote CH$_4$ generation in our work. As is widely reported, the rate-determining step for

CH$_4$ production is the reaction (Supplementary Table 2)[21,34,35].

$$*CO + *H \rightarrow *CHO \tag{1}$$

Therefore, a higher *CO coverage thermodynamically favors the reaction (Eq. 1) and thus should promote CH$_4$ production. Among the three low-indexed Cu single crystal surfaces (111, 110 and 100), (110) has the strongest *CO binding strength[36]. To probe the interaction between *CO and the reconstructed Cu(110) facets, in-situ attenuated total reflectance-surface enhanced infrared absorption spectroscopy (ATR-SEIRAS) was conducted during the cathodic scan of a poly-Cu-coated Si crystal in a CO$_2$-saturated 0.5 M KHCO$_3$ electrolyte with and without EDTMPA from 0.1 to $-1.1$ V versus RHE. In the vibration region of the linearly-bonded CO absorption band from 2040 to 2080 cm$^{-1}$ (ref. [37]), the spectra in electrolytes with and without EDTMPA are quite different (Fig. 3a). What is first apparent is that the electrode tested with EDTMPA always has a much stronger intensity of the surface-bonded CO peak than the one tested without EDTMPA during the cathodic scanning, as is more obviously seen using the integrated area in Fig. 3b, which shows that the reconstructed Cu(110) facets have much more *CO adsorption sites. When the potential is more negative than $-0.47$ V versus RHE, the integrated area of the CO peaks begins to decrease, which might be caused by either *CO desorption to deliver CO gas or *CO protonation to form *CHO. Another significant feature of the spectra, in the presence of EDTMPA, is that the CO band position has an initial blue shift and a subsequent red shift during cathodic scanning, which is in stark contrast to the EDTMPA-free case where only a red shift occurs at very negative potentials. The red shift is ascribed to the weakened *CO binding energy caused by the increased charge donation due to the potential-dependent electrostatic field in the double layer[38]. The initial blue shift is attributed to the strong adsorption of CO and hence the high *CO coverage on the reconstructed Cu(110) surfaces in the presence of EDTMPA[39].

In-situ Raman and CO temperature-programmed desorption (TPD) measurements were also conducted to confirm the strong interaction of *CO with the generated Cu(110) surfaces. In the Raman spectra, the peaks located at 280 and 360 cm$^{-1}$ are assignable to restricted rotation of adsorbed CO (P1) and Cu−CO stretching (P2), respectively[40,41] (Fig. 3c). As the P2-to-P1 intensity ratio is directly proportionate to the increased surface *CO concentration[40], the EDTMPA-added case has obviously higher P2-to-P1 ratios than the EDTMPA-free one, indicating a higher *CO coverage on the reconstructed Cu(110) surface in the EDTMPA-added case. The CO TPD curve of the poly-Cu tested in the EDTMPA-added electrolyte has a pronounced broad CO desorption peak compared to the one tested in the EDTMPA-free case (Fig. 3d), showing a strong *CO binding energy of the generated Cu(110) surfaces. All the above results confirm that the reconstructed Cu(110) surfaces induced by EDTMPA have a high *CO coverage and facilitate *CO stabilization, agreeing well with a previous study[36].

**Locally high proton-feeding environment formed by EDTMPA.** Although a restructured Cu(110) surface has been shown to have a strong *CO binding ability, the resulting high *CO coverage usually limits the number of *H adsorption sites[42,43], which is in favor of ethanol formation but adverse to the rate-determining step of CH$_4$ generation (Eq. 1)[44,45]. This is the main reason why Cu(110) has rarely been reported to have a high CH$_4$ FE. Therefore, in addition to the high *CO coverage, an adequate proton supply is indispensable to the high CH$_4$ conversion in our work[46–48]. Considering that EDTMPA is a phosphonic acid, its adsorption on Cu(110) might form a local environment that provides a high proton supply.

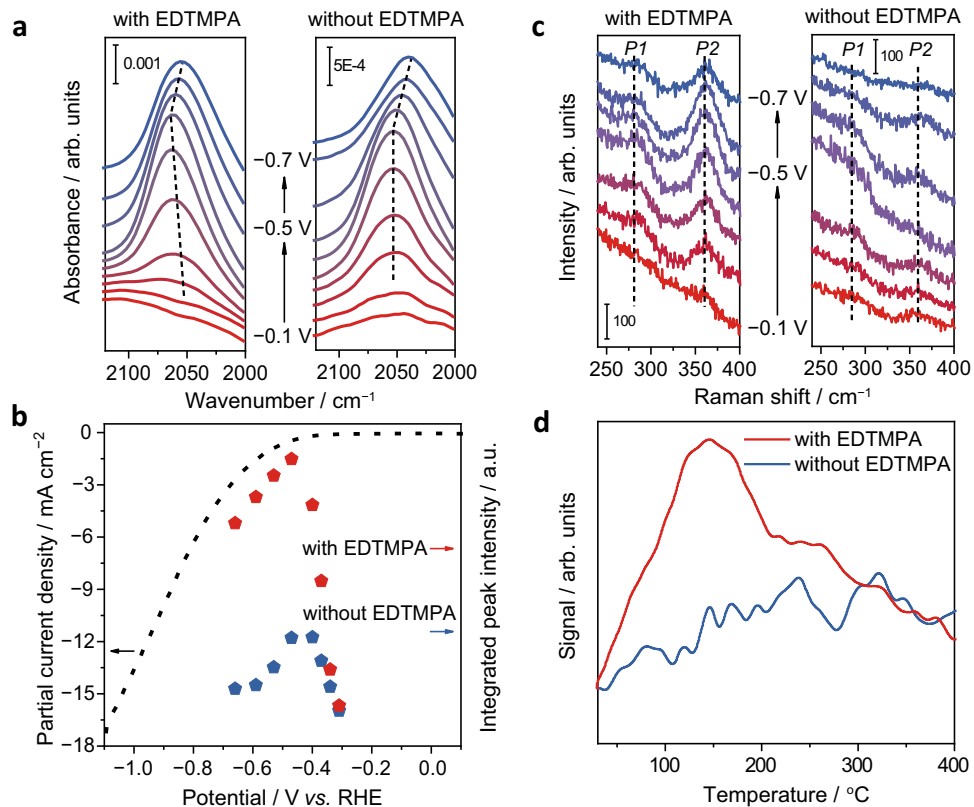

**Fig. 3 Function characterization of Cu(110) surfaces. a** In-situ ATR-SEIRAS spectra of a poly-Cu-coated Si ATR crystal at potentials from −0.1 to −0.7 V versus RHE in $CO_2$-saturated 0.5 M $KHCO_3$ solutions with and without EDTMPA. **b** Cathodic scan curve and comparison of the integrated CO band intensities at potentials from −0.3 to −0.7 V versus RHE as shown in **a**. **c** In-situ Raman spectra of poly-Cu at potentials from −0.1 to −0.7 V versus RHE in $CO_2$-saturated 0.5 M $KHCO_3$ solutions with and without EDTMPA. **d** CO TPD profiles of the poly-Cu electrodes after electrocatalysis in electrolytes with and without EDTMPA.

To explore the function of EDTMPA on this, the $CO_2$ electro-reduction performance was measured in electrolytes containing different amounts of EDTMPA. As the amount of EDTMPA increased from 0 to 8 ppm, both the FEs and partial current densities of $CH_4$ increased while those of $H_2$ remained almost unchanged (Fig. 4a, b). A reasonable interpretation of this is that EDTMPA adsorbed on Cu(110) provides adequate protons to help the protonation of *CO to *CHO rather than the formation of $H_2$ due to the high *CO coverage on Cu(110). When the amount of EDTMPA was increased to 16 ppm, an excessive proton supply resulted in increased $H_2$ production (Fig. 4a, b).

DFT calculations were carried out to explore the role of the adsorbed EDTMPA in increasing the number of protons available for $CH_4$ production. On bare Cu(110), protons are difficult to obtain from $H_2O$ because of the high dissociation barrier (1.32 eV)[49] (Fig. 4c and Supplementary Fig. 14a). However, more protons are available after contacting EDTMPA with Cu(110). In details, one H atom is transferred from *MPA to Cu(110) with an energy decrease of 0.05 eV and a moderate kinetic barrier of 0.73 eV (Fig. 4d and Supplementary Fig. 14b). Subsequently, the *MPA that loses one H (*MPA−H) captures one H atom from the adjacent $H_2O$ molecule to become an *MPA again (Fig. 4d and Supplementary Fig. 15). Note that this process is barrier free with an energy decrease of 0.10 eV, implying that it is both kinetically and thermodynamically favorable. The above DFT results clearly confirm that the adsorbed EDTMPA serves as a proton-delivering medium that accelerates the dissociation of water and continuously provides abundant *H for the conversion of $CO_2$ to $CH_4$.

Apart from its high ability to provide protons, the adsorbed EDTMPA also stabilizes the *CHO species through hydrogen

bond, which is supported by DFT calculations (Fig. 4e and Supplementary Figs. 16, 17). After introducing *MPA on Cu(110), the *CHO binding energy is significantly increased by about 0.44 eV, and the free energy change from *CO to *CHO (Eq. 1) is reduced from 0.79 to 0.41 eV (Fig. 4e), which suggests that the linear-scaling relations between the two coupled intermediates (*CO and *CHO) are broken, thus improving the kinetics of $CH_4$ formation. The improved kinetics were experimentally demonstrated by Tafel analysis with a reduced Tafel slope from 144 to 87 mV dec[−1] in the presence of EDTMPA (Fig. 4f).

**Feasibility study in a flow cell**. Since poor $CO_2$ mass transport through electrolytes significantly limits the current density of the $CO_2RR$ in an H-cell configuration, the scalability of the additive-controlled reconstruction approach was demonstrated in a flow cell configuration for industrial use. The results show that a high $CH_4$ selectivity is still achieved with the EDTMPA additive in an alkaline flow cell, which is totally different from the previous reports of a high $C_2$ selectivity in a pure alkaline electrolyte[23] (Fig. 5a). In a 1 M KOH electrolyte with 13 mM EDTMPA, the poly-Cu gas diffusion electrode (GDE) showed a very stable $CH_4$ FE of 64 ± 2% with a partial current density of 192 ± 6 mA cm[−2] for 6 h (Fig. 5b), and thus a high $CO_2$-to-$CH_4$ conversion rate of 0.25 μmol cm[−2] s[−1] was achieved (Fig. 5c). The high $CH_4$ selectivity indicates that EDTMPA still creates a locally high proton-feeding environment on poly-Cu electrodes even in an alkaline electrolyte, although the conditions were quite different from those in the neutral H-cell. The GIXRD results further illustrate that the surface reconstruction toward Cu(110) is also

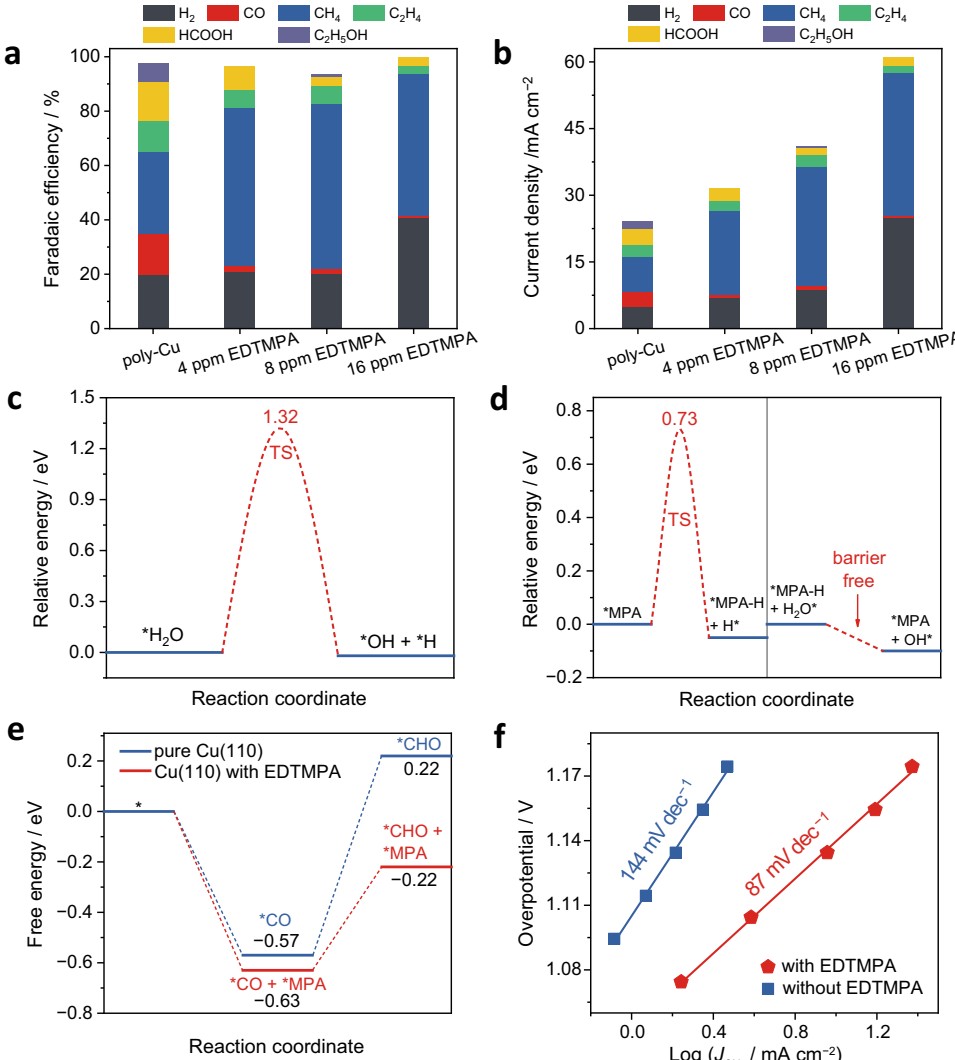

**Fig. 4 Proton-feeding ability of EDTMPA. a, b** Comparison of the FEs (**a**) and partial current densities (**b**) of various $CO_2RR$ products at −1.0 V versus RHE in electrolytes without and with different amounts of EDTMPA (4 ppm, 8 ppm and 16 ppm). **c** Kinetic energy diagram of water dissociation on bare Cu(110). TS stands for the transition state. **d** Kinetic energy diagrams of an H atom transferred from MPA to Cu(110) and then an H atom compensated from $H_2O$ to MPA that loses one H (*MPA−H). **e** Calculated free energy diagrams of the hydrogenation of *CO species to *CHO species at 0 V (versus the standard hydrogen electrode, SHE) on Cu(110) with and without MPA. **f** $CH_4$ Tafel curves in electrolytes with and without EDTMPA.

achieved in the alkaline flow cell (Supplementary Fig. 18), indicating that the additive-controlled reconstruction approach has general applicability.

## Discussion

A controlled surface reconstruction strategy using an EDTMPA additive substantially improves the $CO_2$ electroreduction performance to $CH_4$ on a commercial poly-Cu electrode. The $CO_2$-to-$CH_4$ conversion is promoted by the increased availability of both *H and *CO substrates and stabilization of the *CHO resultant in the rate-determining step of $CH_4$ production (Eq. 1), all of which are resulted from surface reconstruction during the $CO_2RR$. The surface reconstruction involves not only the Cu surface atomic re-arrangement induced by EDTMPA to generate Cu(110) facets, but also the formation of EDTMPA adsorption layer on Cu surface under working potentials.

To illustrate the effects of additives on the surface reconstruction and electrocatalytic performance, we also investigated two analogues of EDTMPA, namely, methylenediphosphonic acid (MDPA) and ethylenediamine tetraacetic acid (EDTA), as electrolyte additives for

the $CO_2RR$. According to SEM and X-ray diffraction (XRD) results, MDPA has an effect on the generation of Cu(110) surface like EDTMPA (Supplementary Fig. 19a, c). However, the $CH_4$ FE suffers from rapid decay accompanied with increased $H_2$ FE (Supplementary Fig. 20a), and their partial current densities are both decreased during one-hour electrocatalysis in the MDPA-added electrolyte (Supplementary Fig. 20b). This may be largely attributed to the smaller space structure of MDPA molecules and the resulting higher coverage on Cu surface compared with EDTMPA, which results in an excessive proton supply and limited $CO_2$ transportation. A substantial stability of the $CO_2$-to-$CH_4$ conversion is achieved after adding EDTA into the electrolyte, although the $CH_4$ FE (~50%) and partial current density (18 mA cm⁻²) are lower than those in the EDTMPA-added case (Supplementary Figs. 1 and 20c, d). This can be attributed to the similar proton-feeding capability of the carboxyl groups in EDTA with the phosphate groups in EDTMPA. The inferior performance for $CH_4$ production is due to the inability of EDTA to induce the atomic re-arrangement of Cu surface to generate Cu(110) like EDTMPA and MDPA (Supplementary Fig. 19b, c). Therefore, it can be concluded that the additives should be rational selected according

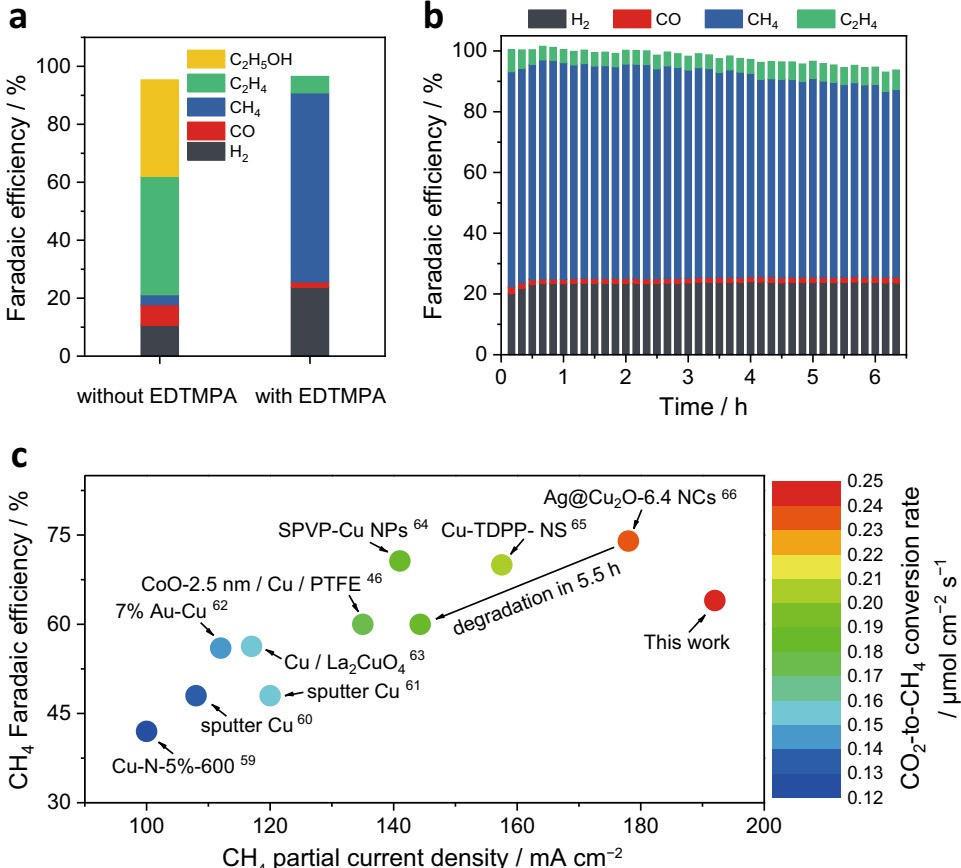

**Fig. 5 CO₂ electroreduction performance in a flow cell. a** Comparison of the FEs of various products at 300 mA cm⁻² on a poly-Cu GDE in 1 M KOH with and without 13 mM EDTMPA additive. **b** Stability test over 6 h of a poly-Cu GDE at 300 mA cm⁻² in 1 M KOH with 13 mM EDTMPA additive. **c** Comparison of our work with previous studies on the electrocatalytic CO₂-to-CH₄ reaction in a flow cell with partial current densities over 100 mA cm⁻² (Refs. [46,59–66]).

to the desired products, although the selecting principles are still under exploration. Incidentally, the fact that the electrolyte effects are noticeable at low concentrations highlights that researchers need to pay careful attention to the presence of additives (intentional) or contaminations (unintentional or unknown) in their experiments.

In this work, EDTMPA is presented as a successful example to illustrate the feasibility of manipulating the surface reconstruction of Cu metal by using selective surface-capping additives to achieve an CO₂ electroreduction performance. Our results show that the additive-controlled surface reconstruction approach has potentially general applicability. Therefore, we believe that more additives will be targetedly developed to facilitate CO₂ electroreduction to other high-value products, and such a strategy will be extended to other chemical process requiring high-performance electrocatalysts.

## Methods

**Chemicals**. KHCO₃ (ACS Reagent 99.7%) was purchased from Sigma Aldrich, and EDTMPA (chelometric titration ≥95.0%) was purchased from TCI. All chemicals were used as received without further purification. Deionized water (18.2 MΩ cm⁻¹) was used throughout all the experiments.

**Preparation of electrodes**. In the H-cell, a commercial Cu mesh (0.5*3.3 cm²) was electropolished in phosphoric acid (85 wt%) at 3 V versus a Cu counter electrode for 180 s, followed by rinsing with deionized water. The cleaned Cu mesh electrode was dried under a N₂ gas flow. In the flow cell, the GDEs were prepared by sputtering 100 nm thick Cu catalysts (Cu target, 99.999%) onto an AvCarb GDS 3250 GDL using a magnetron sputtering system.

**Materials characterization**. TEM was performed using a JEOL JEM-2100F transmission electron microscope operating at 200 kV. For facilitating TEM observation, we electrodeposited a thin layer of nanostructured poly-Cu dendrites

on Cu TEM grids at −15 mA for 3 min in a 0.1 M CuSO₄ electrolyte. SEM was performed on a Hitachi Regulus 8100. XRD patterns were recorded on a Bruker D8 focus diffraction instrument operating at 200 mA and 40 kV. The radiation source was Cu Kα (λ = 0.15418 nm). GIXRD were performed on a Rigaku D/Max 2500 PC diffractometer (Cu-Kα, λ = 1.54056 Å). Elemental analysis was conducted using XPS (Thermo Fisher Scientific K-Alpha⁺, Al Kα radiation, hν = 1486.6 eV). In order to avoid oxidation before ex-situ characterization, all the Cu electrodes after electrocatalysis were immediately washed with N₂-saturated deionized water, following by blow-dry with N₂ as soon as possible, and finally were carefully preserved in a glove box under Ar atmosphere.

**CO TPD**. CO TPD was carried out on a TPDRO apparatus (TP-5080, Tianjin Xianquan Co. Ltd). Prior to the test, Cu electrodes tested with and without EDTMPA were pretreated in a He stream at 150 °C for 1 h with a flow rate of 50 mL min⁻¹ to clean the surface. After cooling to 50 °C under a He atmosphere, the samples were subjected to adsorption of CO for 3 h. The residual CO was removed by purging with He for another 0.5 h. The desorption of CO was then performed by heating at a rate of 10 °C min⁻¹ from 50 to 400 °C, and the TPD signal was recorded using a thermal conductivity detector.

**Electrochemical OH⁻ adsorption**. Electrochemical OH⁻ adsorption was performed in a N₂-saturated 1 M KOH electrolyte using linear sweep voltammetry at a sweep rate of 100 mV s⁻¹ for the poly-Cu electrode. The potential ranged from −0.2 to 0.6 V versus RHE.

**Roughness factor (Rf)**. The roughness factor was estimated from the ratio of the double-layer capacitance (Cdl) between the working electrode and its corresponding smooth Cu foil electrode (assuming that the average double-layer capacitance of a polished smooth Cu electrode is 67 µF cm⁻²)[50], Rf = Cdl/67. Cdl was determined by measuring the capacitive current associated with double-layer charging from the scan-rate dependence of cyclic voltammetric stripping. A series of CV experiments at different scan rates (30–200 mV s⁻¹) were performed in 0.5 M KHCO₃ with/without the EDTMPA additive to calculate Cdl.

**In-situ Raman.** In-situ Raman spectra were obtained on a Raman spectrometer (LabRAM HR spectrometer, Horiba) with a laser wavelength of 785 nm. The CO-adsorption was monitored using a homemade electrolyzer with a $CO_2$-saturated 0.5 M $KHCO_3$ aqueous solution as the electrolyte.

**In-situ ATR-SEIRAS.** Cu-coated Si ATR crystal was used as the working electrode for in-situ ATR-SEIRAS analysis in a $CO_2$-saturated 0.5 M $KHCO_3$ aqueous solution with and without EDTMPA. A Thermo Fisher Nicolet IS50 spectrometer equipped with a MCT detector and a Pike Technologies VeeMAX III ATR accessory was used to collect the spectra. All spectra were acquired with a resolution of 4 cm$^{-1}$ by accumulating eight scans.

**Electrochemical measurements in the H-cell configuration.** Controlled-potential electrolysis was conducted in an H-cell system, which was separated by an anion exchange membrane. An electropolished commercial Cu mesh was used as the working electrode. A graphite rod and a KCl saturated Ag/AgCl electrode were used as the counter and reference electrodes, respectively. The potentials were controlled by an electrochemical working station (IVIUM). All potentials in this study were measured against the Ag/AgCl reference electrode and converted to the RHE reference scale by $E$ (vs. RHE) = $E$ (vs. Ag/AgCl) + 0.197 V + 0.0591 × pH.

The electroreduction of $CO_2$ was conducted in $CO_2$-saturated 0.5 M $KHCO_3$ solution without or with different amounts of EDTMPA additives at room temperature and atmospheric pressure. The electrolyte was purged with $CO_2$ for at least 30 min to remove residual air in the reservoir.

**Electrochemical measurements in the flow cell configuration.** The controlled-current electrolysis was performed in an electrochemical flow cell with a three-electrode system using an electrochemical working station (PARSTAT3000A-DX). 30 mL of 1 M KOH aqueous solution with or without 13 mM EDTMPA additive was circulated through the cathode chamber at a constant rate of 6.4 ml min$^{-1}$ by a peristaltic pump. An anolyte using 1 M KOH was introduced to the anode chamber by a diaphragm pump. An anion exchange membrane (Fumasep FAB-PK-130) was used to separate the cathode and anode chambers. For the $CO_2$RR, gaseous $CO_2$ (99.999%) was passed through the gas chamber at the back side of the Cu GDE (the exposed geometric area was $1 \times 1$ cm$^2$) at a flow rate of 14 ml min$^{-1}$. The $CO_2$ outlet flowrates were recorded by a mass flow detector (Alicat) for accuracy, which was used for all subsequent efficiency calculations. A piece of Ni foam and a Ag/AgCl (saturated KCl) electrode were used as the counter and reference electrodes, respectively. The ohmic loss between the working and reference electrodes was evaluated by electrochemical impedance spectroscopy and 80% iR compensation was applied to correct the potentials manually.

During the reaction, gas phase products were quantified with gas chromatography (SHIMADZU GC-2014), equipped with a thermal conductivity detector and a flame ionization detector. Liquid products were quantified with a 400 MHz $^1H$ NMR spectrometer with water suppression.

The FE of the catalysts were calculated using FE = $\alpha$nF/Q, where $\alpha$ is the number of electrons transferred ($\alpha$ = 8 for $CH_4$ and 2 for $H_2$ production), n is the number of moles for a given product, F is the Faraday constant (96,485 C mol$^{-1}$), Q is all the charge passed throughout the electrocatalysis process (measured by calculating the curve area of current density versus time plot). $CH_4$ and $H_2$ mole fractions were calculated using GC calibration curve.

**Computational methods and models.** All DFT calculations were carried out using the Vienna Ab Initio Simulation Package (VASP)[51]. The exchange correlation energy was represented by the Perdew-Burke-Ernzerhof (PBE) functional (Supplementary Table 3) within the generalized gradient approximation (GGA) and the electron-ion interactions were described by the projector augmented wave (PAW) method[52,53]. The (111) and (100) surfaces were constructed using four atomic layers with a 6×6 supercell, while a $4 \times 6$ supercell (four atomic layers) was used to simulate the (110) surface unless otherwise stated. Only the top two layers were fully relaxed during geometry optimizations. A vacuum layer of 20 Å was set between the periodically repeated slabs along the z direction, and the Brillouin zone was sampled by a Monkhorst-Pack k-point mesh of $3 \times 3 \times 1$. The plane-wave cutoff energy was set to 500 eV, and the convergence criteria for force and energy difference were 0.05 eV/Å and 10$^{-5}$ eV, respectively. Dipole correction was applied to correct potential spurious terms a caused by the asymmetry of the slabs[54,55]. To take into account the van der Waals (vdW) interactions, the empirical correction in Grimme's method (DFT + D3) was used[56]. The minimum energy paths and saddle points were conducted with the climbing image nudged elastic band (CI-NEB) method[57].

The computational hydrogen electrode (CHE) model[58] was used to calculated the Gibbs reaction free energy change ($\Delta G$) of each step. The chemical potential of the proton-electron pair in an aqueous solution is related to one-half of the chemical potential of an isolated hydrogen molecule. In this model, the $\Delta G$ value can be obtained by the formula: $\Delta G = \Delta E + \Delta ZPE - T\Delta S$, where $\Delta E$ is the reaction energy of the reactant and the product species adsorbed on the catalyst, which was directly obtained from DFT calculations; $\Delta ZPE$ and $\Delta S$ are the changes in zero point energies and entropy at 298.15 K, which were calculated from the vibrational frequencies.

The adsorption energy ($E_{ads}$) of the MPA ($CH_5O_3P$, one leg of the EDTMPA molecule) on the (100), (110) and (111) surfaces of Cu was calculated based on the equation: $E_{ads} = E_{total} - E_{substrate} - E_{adsorbate}$, where $E_{total}$, $E_{substrate}$ and $E_{adsorbate}$ are the total energies of the systems containing the substrate and adsorbate, the substrate, and the adsorbate, respectively. According to this definition, a more negative adsorption energy indicates a stronger adsorption.

## Data availability

The data that support the findings of this study are available in the online version of this paper and the accompanying Supplementary Information, or available from the corresponding authors on reasonable request.

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

## Acknowledgements
We gratefully appreciate the support of the National Natural Science Foundation of China (No. 51972223, 51972312 and 22109116), the Natural Science Foundation of Tianjin (No. 20JCYBJC01550) and the Haihe Laboratory of Sustainable Chemical Transformations. The theoretical calculations in this work were performed on TianHe-1(A) at the National Supercomputer Center in Tianjin and Tianhe-2 at the National Supercomputer Center in Guangzhou.

## Author contributions
Z.W. proposed the project, Z.W., Z.H., D.H. conceived the idea, and Z.W., L.Y., Q.Y. supervised the project. Z.H. performed the characterizations and electrochemical measurements. Z.C. and L.Y. performed the DFT calculations and data analysis. Z.H., D.H., J.G., G.J., X.W., S. Lyu., Y.G. and C.G. contributed to the structural and performance analysis. Z.H., D.H., Z.C., Z.W., L.Y. and Q.Y. organized and wrote the manuscript. All authors contributed to the discussion and revision of the manuscript at all stages. Z.H., D.H. and Z.C. contributed equally to this work.

## Competing interests
The authors declare no competing interests.
