## [Peer review file · Nature Communications]

REVIEWER COMMENTS

Reviewer #1 (Remarks to the Author):

In this manuscript by Han et al., the authors describe how electrolyte additives, like a large phosphonic acid, steer the reconstruction of copper electrodes during the CO₂ electroreduction reaction. The authors infer that the additive stabilizes (110) surfaces, as analyzed by structural characterization and DFT calculations, which in turn stabilize *CO intermediates for methane formation. Furthermore, the authors used in situ vibrational spectroscopy to test this hypothesis, and also showed the feasibility of their approach under industrial conditions in an alkaline electrolyzer. Taken together, I think the findings in this manuscript are interesting and original, and therefore would merit publication in Nature Communications, but I still have some questions for the authors that might improve the manuscript further. After these minor revisions are taken into account, I would be happy to review a revised version of the manuscript for publication.

1. How long were the FE measurements in Figure 1a conducted?
2. The FEs in Figure 1c at early times are the same, whereas in the main text only the average FEs are mentioned (without error bars!).
3. I would like to see a more extended analysis of the HRTEM data in Figure 2c, in order to conclude that most of the surface is indeed (110) terminated. Maybe STEM imaging would be beneficial for this as well.
4. The rationale for the large phosphonic acid molecule is not clear to me. What happens for smaller phosphonic acids, like MPA used in the calculations?
5. The discussion about methane formation on (111) vs (110) could be extended in my view. Consensus in literature is that (111) forms methane, and recent studies claim ethanol formation on (110). Furthermore, what is the coverage of EDTMPA on (110)? Would it not block all CO₂RR activity?
6. The hypothesis of higher *CO coverage is interesting, and could be confirmed by analyzing the Raman spectra at low Raman shift. Recent work showed that the 280 and 350 cm⁻¹ peak ratio is related to *CO coverage (ACS Catal. 2021, 11, 13, 7694–7701)
7. How is EDTMPA coordinated at the surface experimentally? This could also be confirmed by Raman measurements at low shifts (Cu-O vibration).
8. The observation of an initial blue shift of *CO in the presence of EDTMPA is also interesting. The discussion could be extended in my view. Does this mean the *CO is more reactive in this case, as was inferred recently? (Angew. Chem. Int. Ed., 2021, 60, 16576)
9. Changes in peak intensity in Raman spectroscopy could also be related to variations in surface enhancement (SERS) and cannot be interpreted on a quantitative basis (page 11, line 203)

10. What is the missing FE in Figure 4a? Formate?
11. The hypothesis of high *CO coverage was also used to explain C-C coupling, why does it promote CH₄ now? The discussion could be elaborated in my view.
12. The CVs in SI figure 8 seem to show reduction waves in the EDTMPA electrolyte, could the authors comment on that?
13. Why does EDTMPA selectively adsorb on a negatively charged surface?
14. Typo on page 15, line 277: DTF instead of DFT
15. What happens after the 6h of testing in Figure 5b?
16. Although the title is nice and catchy, it does not capture everything that is presented in the manuscript, which is mainly about the electrolyte additive. I suggest to add 'with electrolyte additives' to the title.

Reviewer #2 (Remarks to the Author):

The author controlled the reconstruction of Cu surface toward Cu(110) by the introduction of EDTMPA, thus facilitating the selective electroreduction of CO₂ to CH₄. The mechanism of CO₂-to-CH₄ conversion in the presence of EDTMPA was elaborated in detail by a combination of electrochemical experiments, characterization and DFT calculations. SEM, TEM, XRD and hydroxide electroadsorption technique confirmed the ability of EDTMPA to reconstruct the poly-Cu electrode toward Cu(110). DFT results further demonstrated that the introduction of EDTMPA stabilized the Cu(110) surface, making it more likely to be present in the reaction. The results from ATR-SEIRAS, in-situ Raman and TPD measurements showed that the coverage of CO was enhanced with EDTMPA-added electrolyte, which facilitated the rate-limiting step of CO₂-to-CH₄. DFT calculation indicated that the role of EDTMPA is, firstly providing protons for the generation of CHO*, and secondly stabilizing the CHO* through hydrogen bonds. Overall, a record CO₂-to-CH₄ conversion rate was achieved in a flow cell among existing Cu-based studies. There are some concerns needed to be addressed before publication.

1. Main concerns: The author stated the adsorbed EDTMPA stabilized Cu(110) and thus Cu(110) surfaces are preferentially generated during electrocatalysis. However, the adsorption energy obtained from DFT cannot draw a conclusion about the stabilizing effect. If EDTMPA has stronger adsorption energy over the Cu(110) surface rather than the (100) and (111) surfaces, that indicates EDTMPA will most likely poison the surface instead of stabilizing the surface. The DFT simulations here and the experiment results are conflicting.

2. The author mentioned blue shift and red shift and attribute them to Stark tuning effect, donor-acceptor charge transfer and dipole-dipole coupling effect. The explanation is poorly described here. For example, rather than citing a reference to talk about the IR, Roman shift, the DFT should provide correspondingly vibrational frequency calculation of CO. Secondly, a clear sign in the figure is needed to talk about which effect contributes to which shift. Also, it is insufficient to just simply cite some literature to talk about dipole-dipole interaction, Stark tuning effect, donor-acceptor effect. A more comprehensive and specific with a combination of DFT calculation explanation was supposed to be given.

3. How authors ensure the potential rate-limiting step of CO₂-to-CH₄ is CO+H → CHO? Instead of CO+H → COH. Based on the literature, ACS Catal. 2018, 8, 5240–5249, the potential rate-limiting step should be CO+H → COH under electrocatalytic potential.

4. PBE function in DFT calculations couldn't capture CO binding energy and its adsorption site correctly. Please use RPBE function to confirm your DFT calculations.

5. It is better to show all the species involved in the reactions in Fig(4e) like what have been done in Fig4(c and d). Otherwise the readers might be confused about the inconsistent elements before and after the reaction.

6. In line 277, there is a typo 'DTF'. In addition, the author suggested that the *CHO binding energy is increased and the free energy is reduced through hydrogen bond. However, firstly, the *CHO binding energy with and without EDTMPA were not calculated; secondly, there isn't adequate evidence that hydrogen bond leads to this favorability of *CHO.

Reviewer #3 (Remarks to the Author):

The manuscript by Han et al. examines how a specific additive (ethylenediamine tetramethylenephosphonic acid) impact surface reconstruction and catalytic performance of copper electrodes in the electrochemical CO₂ reduction. The novelty of the work derives from the fact that the authors recognize the critical role of surface reconstruction during catalysis, but instead of avoiding reconstruction, they embrace it as an opportunity to deliberately tune surface reconstruction via addition of a surface active additive. Moreover, the fact that the authors go beyond the electrochemical, structural analysis and DFT simulations to include flow cell measurements is noted as a strength. As such, the paper has a well-defined hypothesis, i.e. deliberately steering surface reconstruction via additives in the electrolyte.

The authors present electrochemical analysis (Faradaic efficiency, partial current density) to contrast the specific case of with and without EDTMPA additive. Electrochemical measurements are complemented by structural analysis (SEM, TEM, XRD) to test the interpretation that EDTMPA stabilizes the (110) surface. The authors also provide DFT calculations to compare EDTMPA binding energy on 111, 100, and 110 surfaces. ATR-SEIRAS measurements support the interpretation that the generated (110) surfaces (in the presence of EDTMPA) aid to provide higher *CO coverage and stabilization.

The authors provide a thorough discussion of the experimental results and computational modeling in support of the hypothesized role of EDTMPA. This comprehensive analysis leads the authors to conclude that the additive plays three key roles: (1) facilitate surface reconstruction to favor 110 facets, (2) the involvement of EDTMPA in mediating a proposed proton-feeding mechanism, and (3) stabilization of *CHO via hydrogen bonding. Overall, the results are interesting, but before the manuscript can be recommended for publication in Nature Communications there are several issues (detailed below) which need to be addressed.

1. Experimental analysis of surface reconstruction is challenging; this is typically done with electrochemical scanning tunneling microscopy. Whereas other recent studies (vide infra) have shown agreement between X-ray scattering and STM analysis of surface reconstruction, there are several aspects and potential caveats in the structure characterization that need to be taken into consideration.
2. The facile oxidation of copper electrode surfaces presents a challenge when correlating ex-situ structure analysis with in-situ catalytic performance. Can the authors comment on the possible role of surface oxidation and the impact on structure analysis?
3. The SEM images provided in the main text and the supporting information show that the presence of the EDTMPA additive appears to impact the morphology of the surface, but it's difficult to interpret these images in support of the hypothesized addition of more (110) facets with EDTMPA. Both surfaces undergo macroscopic morphology change, but the SEM images are inadequate to unambiguously support this. The features shown in the images are 10s of nm, from which one can't make direct interpretations of the surface construction or enhanced presence of 110 facets.
4. The TEM image of a copper TEM grid doesn't add much to support the electrochemical measurements or the connection between the hypothesized surface reconstruction and catalytic activity. As the Cu TEM grids are polycrystalline, one can scan around the sample to find a variety of edge faceting, so picking an image that shows a grain with a (110) surface surrounded by other (hkl) surfaces doesn't add much.
5. The XRD patterns in Figure 2d show that the film treated with EDTMPA shows stronger reflection of the (110) planes, with a stronger (110)/(111) ration than the bulk (and presumably the powder reference pattern which should also be included). However, one has to be careful with the XRD analysis, to really be sure that the observed change in reflection ratios is a surface effect, the XRD measurements should be done in grazing incidence configuration (see e.g., Scott et al. ACS Energy Letters 2019, 4 (3), 803-804. DOI: 10.1021/acsenerylett.9b00172)
6. The authors hypothesize a mechanistic explanation that includes three main effects of the additive on the observed catalytic performance: (1) surface reconstruction, (2) mediating proton-feeding, and (3)

*CHO stabilization. Whereas the paper provides a thorough analysis of the effect of EDTMPA, there is a missed opportunity to enhance the impact and shared insights of this paper by connecting the hypothesized mechanism with the underlying molecular interactions of the EDTMPA and the copper surface.

- a. What is it about EDTMPA that enables these unique interactions?
- b. Would this effect also hold for other phosphonic acids or the carbonate analog (EDTA)?
- c. The fact that this effect is pronounced at such low concentrations (8ppm) raises questions about the role of other additives (intentional) or contaminations (unintentional and unknown). Can the authors comment on the possible role / presence of other surface active species in the electrolyte?

Minor point: The abstract states " Electrocatalytic CO₂ reduction to value-added hydrocarbon products using metallic copper (Cu) catalysts is a *sustainable approach to facilitate carbon neutrality. " ; the reviewer suggests revising this as 'a potentially sustainable' approach as there still persist several hurdles to translate scientific discoveries to a truly sustainable, and deployable (scalable, economically viable etc.) technology

Response to the reviewers' comments

Reviewer 1:

General Comments R1: *In this manuscript by Han et al., the authors describe how electrolyte additives, like a large phosphonic acid, steer the reconstruction of copper electrodes during the CO₂ electroreduction reaction. The authors infer that the additive stabilizes (110) surfaces, as analyzed by structural characterization and DFT calculations, which in turn stabilize *CO intermediates for methane formation. Furthermore, the authors used in situ vibrational spectroscopy to test this hypothesis, and also showed the feasibility of their approach under industrial conditions in an alkaline electrolyzer. Taken together, I think the findings in this manuscript are interesting and original, and therefore would merit publication in Nature Communications, but I still have some questions for the authors that might improve the manuscript further. After these minor revisions are taken into account, I would be happy to review a revised version of the manuscript for publication.*

Reply: We thank the reviewer very much for endorsing our key findings and raising insightful comments. All the concerns have been considered seriously and addressed in the revised manuscript. We hope that the reviewer kindly finds the revised manuscript suitable for publication now. Please see below our point-by-point responses.

Specific Comment R1-1: *How long were the FE measurements in Figure 1a conducted?*

Reply: We thank the reviewer for this comment. The duration of the electrolysis (1 h) has been added in the revised manuscript (Line 102, Page 6).

“..... during one-hour tests.”

Specific Comment R1-2: *The FEs in Figure 1c at early times are the same, whereas in the main text only the average FEs are mentioned (without error bars!).*

Reply: We thank the reviewer for pointing out this. Error bars have been provided in Fig. 1a, b in the revised manuscript.

Fig. 1 | CO₂ electroreduction performance in an H-cell. a, b, Comparison of average FEs (a) and partial current densities (b) of various products at -1.0 V versus RHE in the electrolytes with and without EDTMPA during one-hour tests. The error bars in a and b represent mean absolute deviation. c, d, Stability of FEs (c) and partial current densities (d) of the CO and CH₄ products in the electrolytes with and without EDTMPA.

Specific Comment R1-3: *I would like to see a more extended analysis of the HRTEM data in Figure 2c, in order to conclude that most of the surface is indeed (110) terminated. Maybe STEM imaging would be beneficial for this as well.*

Reply: We thank the reviewer for this valuable suggestion. We have provided more TEM images evidencing the terminated surface of (110) plane in Fig. 2b, c and Supplementary Figs. 4–6. We have also added related descriptions in the revised manuscript (Line 132–144, Page 8).

“..... we used electrodeposited Cu TEM grids as poly-Cu electrodes for the CO₂RR and probed their crystal structure before and after electrocatalysis. Representative TEM images of the Cu grid after electrocatalysis in the EDTMPA-added electrolyte show a lot of Cu nanocrystals with hexagonal and cubic outlines, in good agreement with the ideal projections of a rhombic dodecahedral model bounded by {110} facets from different directions²⁷ (Fig. 2b and Supplementary Fig. 4). The Cu rhombic dodecahedrons are further confirmed by the high-resolution TEM (HRTEM) image and the corresponding selected area electron diffraction (SAED) pattern of the equilateral hexagonal projection shape of Cu nanocrystals along [111]²⁷ (Fig. 2c). However, irregular Cu nanoparticles without any preferential surface orientation are merely observed for the electrodeposited Cu TEM grids before and after electrocatalysis in the EDTMPA-free electrolyte (Supplementary Figs. 5, 6).”

Fig. 2 | Characterization of the poly-Cu electrodes in different electrolytes. **a**, SEM image of a poly-Cu electrode after electrocatalysis in the EDTMPA-added electrolyte. **b**, **c**, TEM image (**b**) and HRTEM image (**c**) of an electrodeposited Cu TEM grid after electrocatalysis in the EDTMPA-added electrolyte. Inset in **c** shows the corresponding SAED pattern. **d**, GIXRD patterns of poly-Cu electrodes before/after electrocatalysis in electrolytes with and without EDTMPA. **e**, OH⁻ electroadsorption profiles on the poly-Cu electrodes after electrocatalysis in electrolytes

with and without EDTMPA at a sweep rate of 100 mV s^{-1} in 1 M KOH. **f**, DFT-calculated adsorption energies of an MPA molecule on Cu(110), Cu(100) and Cu(111).

Supplementary Fig. 4 | TEM image of an electrodeposited Cu TEM grid after electrocatalysis in the electrolyte with EDTMPA.

Supplementary Fig. 5 | TEM characterization of the as-electrodeposited Cu TEM grid. a, b, TEM images. c, The corresponding SAED pattern of b.

Supplementary Fig. 6 | TEM characterization of an electrodeposited Cu TEM grid after electrocatalysis in the electrolyte without EDTMPA. a, b, TEM images. c, The corresponding SAED pattern of b.

Specific Comment R1-4: *The rationale for the large phosphonic acid molecule is not clear to me. What happens for smaller phosphonic acids, like MPA used in the calculations?*

Reply: We thank the reviewer for the insightful comment. Since MPA is not in stock currently, we selected another small phosphonic acid, methylenediphosphonic acid (MDPA), a two phosphonic counterpart of MPA. We conducted both electrochemical tests and structural characterizations as shown in the Supplementary Fig. 19a, c and Supplementary Fig. 20a, b in the revised Supplementary Information. The SEM and XRD results showed that reconstruction toward (110) facets also took place with the addition of 8 ppm MDPA, indicating the selective surface-capping effect of phosphonic acid molecules on the formation of Cu(110) surface. For the electrochemical performance, the initial high CH₄ FE (50%) and partial current density (16 mA cm⁻²) suffered from rapid decay accompanied with increased HER FE (Supplementary Fig. 20a, b), indicating MDPA's promotional effect on H₂ production. The CH₄ activity decline with MDPA additive may be due to the smaller skeleton structure of the MDPA molecule and the resulting higher surface coverage per unit compared with EDTMPA, which could provide excessive *H species for HER and limit CO₂ transportation (insufficient steric hindrance). We have provided the relevant discussion in the revised manuscript (Line 349–360, Page 20).

“To illustrate the effects of additives on the surface reconstruction and electrocatalytic performance, we also investigated two analogues of EDTMPA, namely, methylenediphosphonic acid (MDPA) and ethylenediamine tetraacetic acid (EDTA), as electrolyte additives for the CO₂RR. According to SEM and X-ray diffraction (XRD) results, MDPA has an effect on the generation of Cu(110) surface like EDTMPA (Supplementary Fig. 19a, c). However, the CH₄ FE suffers from rapid decay accompanied with increased H₂ FE (Supplementary Fig. 20a), and their partial current densities are both decreased during one-hour electrocatalysis in the MDPA-added electrolyte (Supplementary Fig. 20b). This may be largely attributed to

the smaller space structure of MDPA molecules and the resulting higher coverage on Cu surface compared with EDTMPA, which results in an excessive proton supply and limited CO₂ transportation.”

Supplementary Fig. 19 | Characterization of the poly-Cu electrodes with different additives. a, b, SEM images of poly-Cu electrodes after electrocatalysis in the electrolyte with MDPA (a) and EDTA (b). c, XRD patterns of poly-Cu electrodes after electrocatalysis in the electrolytes with MDPA and EDTA.

Supplementary Fig. 20 | CO₂ electroreduction performance with different additives. a, b, FEs (a) and partial current densities (b) of various products at -1.0 V versus RHE in the electrolytes with 8 ppm MDPA. c, d, FEs (c) and partial current densities (d) of various products at -1.0 V versus RHE in the electrolytes with 8 ppm EDTA.

Specific Comment R1-5: *The discussion about methane formation on (111) vs (110) could be extended in my view. Consensus in literature is that (111) forms methane, and recent studies claim ethanol formation on (110). Furthermore, what is the coverage of EDTMPA on (110)? Would it not block all CO₂RR activity?*

Reply: We thank the reviewer for this constructive comment. Indeed, recent works have revealed that C₂ products will be obtained on Cu(110) surface (*J. Mol. Catal. A-Chem*, 2003, 199, 39–47; *Chem. Rev.*, 2019, 119, 7610–7672). However, in our case, the CH₄ was found to be the main product during the CO₂RR process, which can be attributed to the following reasons. After adding the EDTMPA molecules to the system, an H-rich environment near Cu(110) surface can be achieved, which could promote the hydronation of adsorbed CO molecules to CH₄ product. In this regard, we further calculated the kinetic pathway of hydrogen atom transferred from the surface to adsorbed CO to form *CHO intermediate by DFT calculations, as shown in Fig. R1. A moderate kinetic energy barrier of 0.75 eV can be observed during this transition, which is significantly lower than that of C–C coupling process (1.73 or 1.96 eV) on pure Cu(110) surface (*ACS Catal.*, 2017, 7, 1749–1756). This result clearly indicates that under the hydrogen-rich condition, the CO₂RR will preferentially proceed towards the CH₄ product. We have added a related description in the revised manuscript (Line 265, Page 15).

“..... is in favour of ethanol formation”

Fig. R1 | Illustration of the kinetic process for the formation of *CHO intermediate. The calculated kinetic pathway of hydrogen atom transferred from the surface to adsorbed CO to form *CHO intermediate, with the corresponding atomic structures of initial state (IS), transition state (TS) and final state (FS).

Regarding to the coverage of EDTMPA on Cu(110), we first structurally relaxed a complete EDTMPA molecule on Cu(110) surface with 2×3 supercell (Supplementary Fig. 13a), which represents the extreme cases, that is, the surface is fully and periodically covered by EDTMPA molecules (Supplementary Fig. 13b). Therefore, the maximum coverage of EDTMPA is 1/6 on Cu(110). Obviously, despite such extreme assumptions, an adsorbed EDTMPA molecule only occupies two surface Cu atoms per six ones, and the other four Cu atoms are unoccupied (marked with green circles in Supplementary Fig.13a, b). Furthermore, we explored the transfer process of a CO₂ molecule from the outside of the adsorbed EDTMPA molecule layer to its interior. As is presented in Supplementary Fig. 13c, d, an exothermic process can be observed, showing that CO₂ molecules can approach the Cu(110) surface without obvious obstacles. Therefore, a conclusion can be drawn that the EDTMPA molecules do not block all CO₂ activity. To clearly demonstrate the coverage of EDTMPA molecules on Cu(110), we have added additional results in the revised Supplementary Information (Supplementary Fig. 13). At the same time, we have added relevant descriptions in the revised manuscript (Line 198–208, Page 11).

“The coverage of EDTMPA on Cu(110) was also considered to rule out their possibility of blocking active sites for the CO₂RR. A complete EDTMPA molecule was structurally relaxed on Cu(110) surface with 2×3 supercell, which represents the close-packed structure of EDTMPA molecules on Cu(110) (Supplementary Fig. 13a, b). Obviously, an adsorbed EDTMPA molecule only occupies up to two Cu atoms per six ones, and the other Cu atoms are available for the CO₂RR (marked with green circles in Supplementary Fig. 13a, b). Furthermore, we explored the transfer process of a CO₂ molecule through an EDTMPA adsorption layer on Cu(110) (Supplementary Fig. 13c, d). DFT results show that it is an exothermic process, indicating that CO₂ molecules can approach the Cu(110) surface without obstacle. Therefore, the adsorbed EDTMPA molecules do not block the CO₂RR.”

Supplementary Fig. 13 | Illustration of the coverage of EDTMPA molecules. a, Atomic structure of a single EDTMPA molecule adsorbed on Cu(110) surface with 2×3 supercells. **b,** Periodic display of a with different views. Note that the unoccupied Cu sites are marked with green circles. **c,** Atomic structure of CO₂ molecule with different locations (configuration 1–4) from above EDTMPA to its interior. Note that the CO₂ molecules are highlighted with blue dashed circles for identification. **d,** Calculated relative energy with regard to the distance (configuration 1–4). The

calculated total energy of configuration 1 was used as a reference (set to be zero), and the distance was defined as the distance between the carbon atoms of the CO₂ molecule and the Cu(110) surface.

Specific Comment R1-6: *The hypothesis of higher *CO coverage is interesting, and could be confirmed by analyzing the Raman spectra at low Raman shift. Recent work showed that the 280 and 350 cm⁻¹ peak ratio is related to *CO coverage (ACS Catal. 2021, 11, 13, 7694–7701)*

Reply: We thank the reviewer for the constructive suggestion. We have carefully read the referred literature (ACS Catal., 2021, 11, 7694–7701) and examined our original Raman spectra. The Raman peaks located at 280 and 355–370 cm⁻¹ did exist. According to the literature, the P2/P1 intensity ratio is a valid measure of the surface coverage of CO and the peak ratio increases with increasing CO concentration at –0.52 V_{RHE}. In our added Raman results as shown in Fig. 3c, the P2/P1 peak ratio in EDTMPA-added electrolyte is obviously higher than the additive-free case at –0.5 V, confirming the higher *CO coverage with additive. We have added related descriptions in the revised manuscript (Line 241–246, Page 13).

“In the Raman spectra (Fig. 3c), the peaks located at 280 and 360 cm⁻¹ are assignable to restricted rotation of adsorbed CO (P1) and Cu-CO stretching (P2), respectively^{40, 41} (Fig. 3c). As the P2-to-P1 intensity ratio is directly proportionate to the increased surface *CO concentration⁴⁰, the EDTMPA-added case has obviously higher P2-to-P1 ratios than the EDTMPA-free one, indicating a higher *CO coverage on the reconstructed Cu(110) surface in the EDTMPA-added case.”

Fig. 3 | Function characterization of Cu(110) surfaces. **a**, ATR-SEIRAS spectra of a poly-Cu-coated Si ATR crystal at potentials from -0.1 to -0.7 V versus RHE in a CO_2 -saturated 0.5 M KHCO_3 solution with and without EDTMPA. **b**, Cathodic scan curve and comparison of the integrated CO band intensities at potentials from -0.3 to -0.7 V versus RHE as shown in **a**. **c**, *In-situ* Raman spectra of poly-Cu at potentials from -0.1 to -0.7 V versus RHE in a CO_2 -saturated 0.5 M KHCO_3 solution with and without EDTMPA. **d**, CO TPD profiles of the poly-Cu electrodes after electrocatalysis in electrolytes with and without EDTMPA.

Specific Comment R1-7: *How is EDTMPA coordinated at the surface experimentally? This could also be confirmed by Raman measurements at low shifts (Cu-O vibration).*

Reply: We thank the reviewer for this helpful comment. We carefully checked the Raman bands at $600\text{--}620\text{ cm}^{-1}$, which have been assigned to Cu–O stretching mode in previous studies (*J. Am. Chem. Soc.*, 2020, 142, 9735–9743). Apparently, the Cu–O vibration is stronger in the presence of EDTMPA additive than the additive-free case as indicated in Supplementary Fig. 12. Therefore, the *in situ* Raman result verified the coordination structure of EDTMPA on the Cu surface, that is

through ‘O’ of –OH groups in EDTMPA, the same as revealed by DFT calculations. We have added the related Raman results in the revised Supplementary Information (Supplementary Fig. 12), with relevant discussion added in the revised manuscript (Line 189–197, Page 11).

“It is indicated that EDTMPA has a potential-driven specific adsorption behavior on the Cu surface³⁰, which was further confirmed by *in-situ* Raman measurement (Supplementary Fig. 12). The Cu–O vibration bands ($600\text{--}620\text{ cm}^{-1}$)³¹ are present in the EDTMPA-added case, and the blue shift occurs as the cathodic polarization increases, indicating that the EDTMPA molecules coordinate at the Cu surface via Cu–O bonds and the bonding strength is proportional to the cathodic polarization, the same as revealed by DFT calculations (Supplementary Figs. 7, 8 and Supplementary Table 1).”

Supplementary Fig. 12 | *In situ* Raman spectra demonstrating the adsorption of EDTMPA on poly-Cu. *In-situ* Raman spectra of poly-Cu at potentials from –0.1 to –0.7 V versus RHE in a CO₂-saturated 0.5 M KHCO₃ solution with and without EDTMPA. The Raman bands at 600–620 cm⁻¹ correspond to Cu–O stretching mode.

Specific Comment R1-8: *The observation of an initial blue shift of *CO in the presence of EDTMPA is also interesting. The discussion could be extended in my view. Does this mean the *CO is more reactive in this case, as was inferred recently? (Angew. Chem. Int. Ed., 2021, 60, 16576)*

Reply: We thank the reviewer for the comment. We carefully read the referred literature (*Angew. Chem. Int. Ed.*, 2021, 60, 16576–16584). The author ascribed 2060 and 2095 cm^{-1} to low-frequency band linear CO (LFB-CO) and high-frequency band linear CO (HFB-CO), respectively. And the highly dynamic peak position shift of the LFB-CO at -0.9 V suggests the active chemical nature of LFB-CO for C–C coupling. However, we did not find any signals near 2060 cm^{-1} in our Raman results, indicating no such reactive *CO sites existed for ethylene production. The initial blue shift of *CO in our work was observed in the *in-situ* attenuated total reflectance-surface enhanced infrared absorption spectroscopy (ATR-SEIRAS) measurement.

Specific Comment R1-9: *Changes in peak intensity in Raman spectroscopy could also be related to variations in surface enhancement (SERS) and cannot be interpreted on a quantitative basis (page 11, line 203)*

Reply: We thank the reviewer for pointing out this issue. It is true that Cu is known to exhibit strong SERS activity and the roughened electrode surface could result in hotspots for enhanced SERS activity (*Angew. Chem. Int. Ed.*, 2021, 60, 16576–16584; *Annu. Rev. Anal. Chem.*, 2008, 1, 601–626). Since the different nanostructured surface of our Cu electrodes with and without electrolyte additive, they could exhibit distinct surface enhancement effect of Raman spectroscopy. Therefore, the quantitative comparison at page 11, line 203 have been deleted for accurate. And the discussions about P2/P1 peak ratio (Fig. 3c) as kindly advised in comment 6, which could evidence *CO coverage variations with and without additive, have been added in the revised manuscript (Line 241–246, Page 13). For details, please refer to our reply to the comment R1-6.

Specific Comment R1-10: *What is the missing FE in Figure 4a? Formate?*

Reply: We thank the reviewer for pointing out this issue. The missing FEs are those of the liquid products (i.e., formate and ethanol), which have been provided in the revised manuscript (Fig. 4a, b).

Fig. 4 | Proton-feeding ability of EDTMPA. **a, b**, Comparison of the FEs (**a**) and partial current densities (**b**) of various CO₂RR products at -1.0 V versus RHE in electrolytes without and with different amounts of EDTMPA (4 ppm, 8 ppm and 16 ppm). **c**, Kinetic energy diagram of water dissociation on bare Cu(110). TS stands for the transition state. **d**, Kinetic energy diagrams of an H atom transferred from EDTMPA to Cu(110) and then an H atom compensated from H₂O to EDTMPA that lost one H (*MPA-H). **e**, Calculated free energy diagrams of the hydrogenation of *CO species to *CHO species at 0 V (versus a standard hydrogen electrode, SHE) on Cu(110) with and without EDTMPA. **f**, CH₄ Tafel curves in electrolytes with and without EDTMPA.

Specific Comment R1-11: *The hypothesis of high *CO coverage was also used to explain C-C coupling, why does it promote CH₄ now? The discussion could be elaborated in my view.*

Reply: We thank the reviewer for this invaluable suggestion. The hydrogen-rich environment induced by the introduction of EDTMPA molecules will promote the hydrogenation of *CO to the C1 product. For details, please refer to our reply to the comment R1-5.

Specific Comment R1-12: *The CVs in SI figure 8 seem to show reduction waves in the EDTMPA electrolyte, could the authors comment on that?*

Reply: We thank the reviewer for this reasonable concern. As shown in the inset in Supplementary Fig. 10, both the EDTMPA-added and EDTMPA-free cases exhibit the similar reduction wave, indicating it is unrelated to the adsorption/desorption or decomposition of EDTMPA. We speculate that the reduction wave indicates the change of dominant proton donors from HCO₃⁻ to H₂O (*ACS Catal.*, 2021, 11, 4936–4945), as suggested by the current plateaued where HCO₃⁻ reduction became limited by mass transport. We have added the enlarged curves as an inset to Supplementary Fig. 10 (Supplementary Fig. 8 in the original submission), with corresponding explanations added below Supplementary Fig. 10.

Supplementary Fig. 10 | The stability of EDTMPA molecules under bias. Comparison of CV curves between 0.3 V and -1.7 V versus RHE at a scan rate of 50 mV s⁻¹ on poly-Cu electrodes in the Ar-saturated 0.5 M KHCO₃ electrolyte with and without 100 ppm EDTMPA.

As shown in the inset, both EDTMPA-added and EDTMPA-free cases exhibit the similar reduction wave, indicating it is unrelated to the adsorption/desorption or decomposition of EDTMPA. We speculate that the reduction wave indicates the change of dominant proton donors from HCO₃⁻ to H₂O¹, as suggested by the current plateaued where HCO₃⁻ reduction becomes limited by mass transport.

Specific Comment R1-13: *Why does EDTMPA selectively adsorb on a negatively charged surface?*

Reply: We thank the reviewer for this comment. We verified the adsorption of EDTMPA by XPS. The N and P signals can only be detected for the electrode after electrocatalysis in the EDTMPA-added electrolyte, but no for the one soaked in the same electrolyte for the same time at an open circuit potential. *In-situ* Raman spectra also show the blue shift of Cu–O vibration bands. The XPS and Raman results indicate that EDTMPA has a potential-driven specific adsorption on Cu surface, and the bonding strength is increased at a cathodic polarization. Furthermore, we have calculated the adsorption energy of a complete EDTMPA molecule on Cu(110) surface under different electric force fields (Supplementary Table 1). Obviously, the applied electric field has a positive effect on the adsorption of EDTMPA. The mechanism for the potential-driven specific adsorption is still not clear. We have added the DFT results in the revised Supplementary Information (Supplementary Table 1), and relevant descriptions in the revised manuscript (Line 189–197, Page 11).

“It is indicated that EDTMPA has a potential-driven specific adsorption behavior on the Cu surface³⁰, which was further confirmed by *in-situ* Raman measurement (Supplementary Fig. 12). The Cu–O vibration bands (600–620 cm⁻¹)³¹ are present in the EDTMPA-added case, and the blue shift occurs as the cathodic polarization increases, indicating that the EDTMPA molecules coordinate at the Cu surface via

Cu–O bonds and the bonding strength is proportional to the cathodic polarization, the same as revealed by DFT calculations (Supplementary Figs. 7, 8 and Supplementary Table 1).”

Supplementary Table 1. The calculated adsorption energy of intact EDTMPA molecule on Cu(110) surface under different electric force fields.

Electric force field (V/Å)	-0.30	-0.20	-0.10	0.00
Adsorption energy (eV)	-2.54	-2.38	-2.25	-2.12

Specific Comment R1-14: *Typo on page 15, line 277: DTF instead of DFT*

Reply: Thank the reviewer for pointing out this issue. We have double-checked the manuscript carefully and corrected the spelling of ‘DFT’ in the revised manuscript (Line 305, Page 17).

Specific Comment R1-15: *What happens after the 6h of testing in Figure 5b?*

Reply: We thank the reviewer for this reasonable suggestion. GIXRD results have been provided in the Supplementary Fig. 18, which illustrate that the surface reconstruction toward (110) facets is also achieved in the alkaline flow cell. We have provided the relevant discussion in the revised manuscript (Line 327–329, Page 18).

“The GIXRD results further illustrate that the surface reconstruction toward Cu(110) is also achieved in the alkaline flow cell (Supplementary Fig. 18),”

Supplementary Fig. 18 | GIXRD patterns of GDEs after electrocatalysis in alkaline electrolytes with and without EDTMPA.

Specific Comment R1-16: *Although the title is nice and catchy, it does not capture everything that is presented in the manuscript, which is mainly about the electrolyte additive. I suggest to add ‘with electrolyte additives’ to the title.*

Reply: We thank the reviewer for this nice suggestion. We have added ‘with electrolyte additives’ to the title in the revised manuscript. The new title is “**Steering surface reconstruction of copper with electrolyte additives for CO₂ electroreduction**”.

Reviewer 2:

General Comments R2: *The author controlled the reconstruction of Cu surface toward Cu(110) by the introduction of EDTMPA, thus facilitating the selective electroreduction of CO₂ to CH₄. The mechanism of CO₂-to-CH₄ conversion in the presence of EDTMPA was elaborated in detail by a combination of electrochemical experiments, characterization and DFT calculations. SEM, TEM, XRD and hydroxide electroadsorption technique confirmed the ability of EDTMPA to reconstruct the poly-Cu electrode toward Cu(110). DFT results further demonstrated that the introduction of EDTMPA stabilized the Cu(110) surface, making it more likely to be present in the reaction. The results from ATR-SEIRAS, in-situ Raman and TPD measurements showed that the coverage of CO was enhanced with EDTMPA-added electrolyte, which facilitated the rate-limiting step of CO₂-to-CH₄. DFT calculation indicated that the role of EDTMPA is, firstly providing protons for the generation of CHO*, and secondly stabilizing the CHO* through hydrogen bonds.*

Overall, a record CO₂-to-CH₄ conversion rate was achieved in a flow cell among existing Cu-based studies. There are some concerns needed to be addressed before publication.

Reply: We thank the reviewer very much for the positive comments and kind valuable suggestions. All the concerns have been considered seriously and addressed in the revised manuscript. We hope that the reviewer kindly finds the revised manuscript suitable for publication now. Please see below our point-by-point responses.

Specific Comment R2-1: *Main concerns: The author stated the adsorbed EDTMPA stabilized Cu(110) and thus Cu(110) surfaces are preferentially generated during electrocatalysis. However, the adsorption energy obtained from DFT cannot draw a conclusion about the stabilizing effect. If EDTMPA has stronger adsorption energy over the Cu(110) surface rather than the (100) and (111) surfaces, that indicates*

EDTMPA will most likely poison the surface instead of stabilizing the surface. The DFT simulations here and the experiment results are conflicting.

Reply: We thank the reviewer for this insightful comment. Firstly, as is revealed by Yang et al. (*Nature*, 2008, 453, 638–641), the adsorbate atom with strong bonding can be used to change the relative stabilities of different crystal facets, and might provide an effective means for stabilizing the surface. In this literature, a high percentage of (001) facets can be obtained when the anatase surfaces are surrounded by F atoms, due to the strong interaction. Similarly, in our case, the MPA molecule (a representative of the EDTMPA molecule) has a higher adsorption energy on Cu(110) (–1.30 eV) than those on Cu(100) (–1.06 eV) and Cu(111) (–1.11 eV), suggesting that the adsorbed EDTMPA molecules will stabilize the intrinsically high-energy Cu(110). Therefore, in a chemical environment rich in EDTMPA molecules, the Cu(110) surface will be generated through electrochemical surface reconstruction.

Regarding to the possibility that EDTMPA molecules with stronger adsorption energy could poison the catalytically active sites, we introduced the extreme case, that is, the surface is fully and periodically covered by EDTMPA molecules, still leaving ~66% of copper atoms to be used for CO₂ reduction (Supplementary Fig. 13a, b). Moreover, CO₂ molecule can approach the Cu(110) surface without any obstacle (Supplementary Fig. 13c, d). Thus, a conclusion can be drawn that the EDTMPA molecules induce the generation of Cu(110) but do not poison the active sites for CO₂RR. We have provided the additional results in the revised Supplementary Information (Supplementary Fig. 13) and added corresponding explanations in the revised manuscript (Line 198–208, Page 11).

“The coverage of EDTMPA on Cu(110) was also considered to rule out their possibility of blocking active sites for the CO₂RR. A complete EDTMPA molecule was structurally relaxed on Cu(110) surface with 2×3 supercell, which represents the close-packed structure of EDTMPA molecules on Cu(110) (Supplementary Fig. 13a, b). Obviously, an adsorbed EDTMPA molecule only occupies up to two Cu atoms per six ones, and the other Cu atoms are available for the CO₂RR (marked with green

circles in Supplementary Fig. 13a, b). Furthermore, we explored the transfer process of a CO₂ molecule through an EDTMPA adsorption layer on Cu(110) (Supplementary Fig. 13c, d). DFT results show that it is an exothermic process, indicating that CO₂ molecules can approach the Cu(110) surface without obstacle. Therefore, the adsorbed EDTMPA molecules do not block the CO₂RR.”

Supplementary Fig. 13 | Illustration of the coverage of EDTMPA molecules. **a**, Atomic structure of a single EDTMPA molecule adsorbed on Cu(110) surface with 2×3 supercells. **b**, Periodic display of **a** with different views. Note that the unoccupied Cu sites are marked with green circles. **c**, Atomic structure of CO₂ molecule with different locations (configuration 1–4) from above EDTMPA to its interior. Note that the CO₂ molecules are highlighted with blue dashed circles for identification. **d**, Calculated relative energy with regard to the distance (configuration 1–4). The calculated total energy of configuration 1 was used as a reference (set to be zero), and the distance was defined as the distance between the carbon atoms of the CO₂ molecule and the Cu(110) surface.

Specific Comment R2-2: *The author mentioned blue shift and red shift and attribute them to Starking tuning effect, donor-acceptor charge transfer and dipole-dipole*

coupling effect. The explanation is poorly described here. For example, rather than citing a reference to talk about the IR, Raman shift, the DFT should provide correspondingly vibrational frequency calculation of CO. Secondly, a clear sign in the figure is needed to talk about which effect contributes to which shift. Also, it is insufficient to just simply cite some literature to talk about dipole-dipole interaction, Stark tuning effect, donor-acceptor effect. A more comprehensive and specific with a combination of DFT calculation explanation was supposed to be given.

Reply: We thank the reviewer for this comment.

The blue shift of CO band position can be explained as a decreased occupancy of π back-bonding orbital with increased *CO coverage, and thus the bond was strengthened and the associated frequency up-shifted (*Surf. Sci.*, 1982, 123, 397–412; *Electrochim. Acta*, 1996, 41, 623–630). The enhanced adsorption of CO in the presence of EDTMPA molecules have been demonstrated by DFT calculations (Fig. 4e). So the dipole-dipole interaction is responsible for the blue shift. On the other hand, the potential-dependent metal-adsorbate chemical bonding induced by the variance of electrostatic field in the double layer, can contribute to the electrochemical Stark effects on the frequencies of adsorbate on electrode surface (*J. Phys. Chem. C*, 2010, 114, 403–411). Specifically, increasing charge donation into the unoccupied $2\pi^*$ CO level lowers the stretching frequency and weakens the bond (*Chem. Phys.*, 1993, 175, 37–51). Therefore, the red shift can be attributed to the vibrational Stark effect.

We need to point out that the DFT study on the vibrational frequency of adsorbed CO molecules under electric field is complex. Also, we have little knowledge on how to explain this phenomenon with reliable and robust DFT calculations. Therefore, we decided to give more digestible explains why the blue/red shift happens, without using any jargons like *starking tuning effect, donor-acceptor charge transfer and dipole-dipole coupling effect*. The explanations have been added in the revised manuscript (Line 234–238, Page 13).

“The red shift is ascribed to the weakened *CO binding energy caused by the increased charge donation due to the potential-dependent electrostatic field in the double layer³⁸. The initial blue shift is attributed to the strong adsorption of CO and hence the high *CO coverage on the reconstructed Cu(110) surfaces in the presence of EDTMPA³⁹.”

Specific Comment R2-3: *How authors ensure the potential rate-limiting step of CO₂-to-CH₄ is CO+H→CHO? Instead of CO+H→COH. Based on the literature, ACS Catal. 2018, 8, 5240–5249, the potential rate-limiting step should be CO+H→COH under electrocatalytic potential.*

Reply: We thank the reviewer for this insightful comment. As mentioned in the literature, the activation energy of *COH formation is significantly lower than that of *CHO formation on Cu(111) surface, thus, the generation of *COH intermediate will be favored at the applied potentials (ACS Catal., 2018, 8, 5240–5249). Similarly, Zhao et al. theoretically revealed that with increasing applied potential, the dominance of *COH (formed via potential-independent surface *H transfer) diminishes, switching to the competitive formation of both *CHO and *COH (both formed via potential-dependent PCET (proton-coupled electron transfer)), eventually, the adsorbed CO reduces almost equally to *COH and *CHO at –0.9 V versus RHE (J. Am. Chem. Soc., 2021, 143, 6152–6164).

In our case of Cu(110) surface, the total energy of *COH intermediate is further calculated to be compared with that of *CHO intermediate, with and without EDTMPA molecule, as summarized in Supplementary Table 2. Clearly, the calculated total energy of *CHO intermediate is significantly lower than that of *COH intermediate, especially in the presence of EDTMPA molecules (the *CHO intermediate can be more stabilized due to hydrogen bonding), proving that the next hydrogenation of CO will undoubtedly form a *CHO intermediate. We have added corresponding explanations in the revised manuscript (Line 212–213, Page 12) and below the Supplementary Table 2 in the revised Supplementary Information.

“As is widely reported, the rate-determining step for CH₄ production is the reaction (Supplementary Table 2)”

Supplementary Table 2. The calculated total energies of *CHO and *COH intermediates on Cu(110) surface and its energy difference, without and with EDTMPA molecule (the unit is eV).

	pure (110) surface	with EDTMPA
*CHO	-357.670	-415.143
*COH	-356.956	-414.116
Energy difference	-0.714	-1.027

Although *CHO is widely known as the intermediate of *CO hydrogenation for CH₄ generation, the generation of *COH intermediate on Cu(111) surface was also reported in some literatures^{7,8}. To confirm the intermediate in our work, the total energies of the two intermediates on Cu(110) surface are calculated with and without EDTMPA molecule (Supplementary Table 2). Clearly, the calculated total energy of *CHO intermediate is significantly lower than that of *COH intermediate, especially in the presence of EDTMPA molecules (the *CHO intermediate can be more stabilized due to hydrogen bonding, as shown in Supplementary Figs. 16, 17), confirming that the rate-determining step for CH₄ production should be *CO + *H → *CHO in our work.

Specific Comment R2-4: PBE function in DFT calculations couldn't capture CO binding energy and its adsorption site correctly. Please use RPBE function to confirm your DFT calculations.

Reply: We thank the reviewer for this reasonable suggestion. Indeed, the revised PBE functional (RPBE) proposed by Nørskov and co-workers, can provide a more accurate description of the adsorption of small molecules, which is closer to the experimental value, effectively avoiding the over-adsorption problem (*Phys. Rev. B*, 1999, 59, 7413–7421).

In this regard, the CO adsorption and hydrogenation on Cu(110) surface via RPBE functional are further calculated to compare with the cases of the PBE functional. As is presented in Table S3, clearly, different functionals have weak effects on CO

adsorption and hydrogenation, proving the rationality and feasibility of our data calculated by the PBE functional. In addition, the PBE functional is also widely used by the theoretical community to explore the CO₂ reduction on Cu-based materials, showing well-established reliability (*J. Am. Chem. Soc.*, 2021, 143, 6152–6164; *J. Am. Chem. Soc.*, 2016, 138, 483–486; *J. Am. Chem. Soc.*, 2017, 139, 130-136; *Angew. Chem. Int. Ed.*, 2013, 125, 2519–2522; *ACS Catal.*, 2019, 9, 6305-6319; *J. Phys. Chem. C*, 2021, 125, 10919–10925). We have added the detailed information to the revised manuscript (Line 483, Page 26) and below the Supplementary Table 3 in the revised Supplementary Information.

“..... represented by the Perdew-Burke-Ernzerhof (PBE) functional (Supplementary Table 3)”

Supplementary Table 3. The calculated adsorption energy of CO molecule ($\Delta E(*CO)$) and energy change for its hydrogenation ($\Delta E(*CO \rightarrow *CHO)$) on Cu(110) surface without and with EDTMPA molecule via PBE and revised PBE (RPBE)⁹ functional, respectively, the unit is eV.

pure (110) surface	$\Delta E(*CO)$	$\Delta E(*CO \rightarrow *CHO)$
PBE functional	-1.10	0.44
RPBE functional	-1.01	0.44

EDTMPA	$\Delta E(*CO)$	$\Delta E(*CO \rightarrow *CHO)$
PBE functional	-1.19	0.10
RPBE functional	-1.04	0.11

The PBE functional is widely used by the theoretical community to explore the CO₂ reduction on Cu-based electrocatalysts, showing well-established reliability¹⁰⁻¹². However, the RPBE functional proposed by Nørskov and co-workers can provide a more accurate description of the adsorption of small molecules, which is closer to the experimental value, effectively avoiding the over-adsorption problem⁹. Here, we compared their calculated results of the CO adsorption and hydrogenation on Cu(110) surface in Supplementary Table 3. It clearly presents that different

functionals have weak effects on CO adsorption and hydrogenation, proving the rationality and feasibility of our data calculated by the PBE functional.

Specific Comment R2-5: *It is better to show all the species involved in the reactions in Fig(4e) like what have been down in Fig4(c and d). Otherwise the readers might be confused about the inconsistent elements before and after the reaction.*

Reply: We thank the reviewer for this nice suggestion. The species involved have been given in Fig. 4e in the revised manuscript.

Fig. 4 | Proton-feeding ability of EDTMPA. **a, b**, Comparison of the FEs (**a**) and partial current densities (**b**) of various CO₂RR products at -1.0 V versus RHE in electrolytes without and with different amounts of EDTMPA (4 ppm, 8 ppm and 16 ppm). **c**, Kinetic energy diagram of water dissociation on bare Cu(110). TS stands for the transition state. **d**, Kinetic energy diagrams of an H atom transferred from EDTMPA to Cu(110) and then an H atom compensated from H₂O to EDTMPA that lost one H (*MPA-H). **e**, Calculated free energy diagrams of the hydrogenation of *CO species to *CHO species at 0 V (versus a standard hydrogen electrode, SHE) on Cu(110) with and without EDTMPA. **f**, CH₄ Tafel curves in electrolytes with and without EDTMPA.

Specific Comment R2-6: *In line 277, there is a typo 'DTF'. In addition, the author suggested that the *CHO binding energy is increase and the free energy is reduced through hydrogen bond. However, firstly, the *CHO binding energy with and without EDTMPA were not calculated; secondly, there isn't adequate evidence that hydrogen bond leads to this favorability of *CHO.*

Reply: We thank the reviewer for these helpful suggestions. We have corrected this typo in our revised manuscript (Line 15, Page 17).

And we have calculated the binding energy of *CHO intermediate with and without EDTMPA in our original manuscript as shown in Fig. 4e. Clearly, this binding energy was enhanced by about 0.44 eV (from 0.22 to -0.22 eV) after introducing the EDTMPA molecule, where the distance between O and H atoms is 1.53 Å (Supplementary Fig. 16), belonging to the range of hydrogen bond (shorter than 3.0 Å) (*Angew. Chem. Int. Ed.*, 2002, 41, 48–76). Furthermore, to convincingly demonstrate the role of hydrogen bond in enhancing the binding strength of the *CHO intermediate, we further calculated the energy-distance relationship between the *CHO intermediate and the adsorbed EDTMPA molecule on Cu(110) surface.

As is shown in Supplementary Fig. 17, a weak adsorption of *CHO intermediate can be observed when it is far away from the EDTMPA molecule, correspondingly, the calculated energy change from *CO to *CHO is basically consistent with that on pure Cu(110) surface without EDTMPA molecule, suggesting no hydrogen bond. Therefore, the above comparison further shows that the hydrogen bond is responsible

for the stabilizing the *CHO intermediate. We have added the corresponding results to the revised Supplementary Information (Supplementary Figs. 16, 17) and described in the revised manuscript (Line 304–306, Page 17) and below Supplementary Fig. 17.

“Apart from its high ability to provide protons, the adsorbed EDTMPA also stabilizes the *CHO species through hydrogen bonds, which is shown by DFT calculations (Fig. 4e and Supplementary Figs. 16, 17).”

Supplementary Fig. 16 | Illustration of hydrogen bonding forming between EDTMPA and *CHO. The corresponding atomic structures of *CO and *CHO intermediate species in a Cu(110) surface without and with EDTMPA molecule. And the distance (1.53 Å) within the range of hydrogen bond (shorter than 3.0 Å) is also marked.

Supplementary Fig. 17 | Illustration of the range of hydrogen bond between EDTMPA and *CHO. The corresponding atomic structures of *CHO intermediate species on the Cu(110) surface with EDTMPA molecule with different distances between EDTMPA and *CHO. And the energy change for *CO hydrogenation to *CHO, i.e., $\Delta E(*CO \rightarrow *CHO)$ with different distances is provided for comparison.

A weak adsorption of *CHO intermediate can be observed when it is far away from the EDTMPA molecule. Correspondingly, the calculated energy change from *CO to *CHO is basically consistent with that on pure Cu(110) surface without EDTMPA molecule, suggesting no hydrogen bond is formed. Therefore, the above comparison indicates that the hydrogen bond is responsible for the stabilizing the *CHO intermediate.

Reviewer 3:

General Comments R3: *The manuscript by Han et al. examines how a specific additive (ethylenediamine tetramethylenephosphonic acid) impact surface reconstruction and catalytic performance of copper electrodes in the electrochemical CO₂ reduction. The novelty of the work derives from the fact that the authors recognize the critical role of surface reconstruction during catalysis, but instead of avoiding reconstruction, they embrace it as an opportunity to deliberately tune surface reconstruction via addition of a surface active additive. Moreover, the fact that the authors go beyond the electrochemical, structural analysis and DFT simulations to include flow cell measurements is noted as a strength. As such, the paper has a well-defined hypothesis, i.e. deliberately steering surface reconstruction via additives in the electrolyte.*

*The authors present electrochemical analysis (Faradaic efficiency, partial current density) to contrast the specific case of with and without EDTMPA additive. Electrochemical measurements are complemented by structural analysis (SEM, TEM, XRD) to test the interpretation that EDTMPA stabilizes the (110) surface. The authors also provide DFT calculations to compare EDTMPA binding energy on 111, 100, and 110 surfaces. ATR-SEIRAS measurements support the interpretation that the generated (110) surfaces (in the presence of EDTMPA) aid to provide higher *CO coverage and stabilization.*

*The authors provide a thorough discussion of the experimental results and computational modeling in support of the hypothesized role of EDTMPA. This comprehensive analysis leads the authors to conclude that the additive plays three key roles: (1) facilitate surface reconstruction to favor 110 facets, (2) the involvement of EDTMPA in mediating a proposed proton-feeding mechanism, and (3) stabilization of *CHO via hydrogen bonding. Overall, the results are interesting, but before the manuscript can be recommended for publication in Nature Communications there are several issues (detailed below) which need to be addressed.*

Reply: We are grateful to the reviewer for endorsing our key findings and raising insightful comments. All the concerns have been considered seriously and addressed in the revised manuscript. We hope that the reviewer kindly finds the revised manuscript suitable for publication now. Please see below our point-by-point response to your concerns.

Specific Comment R3-1: *Experimental analysis of surface reconstruction is challenging; this is typically done with electrochemical scanning tunneling microscopy. Whereas other recent studies (vide infra) have shown agreement between X-ray scattering and STM analysis of surface reconstruction, there are several aspects and potential caveats in the structure characterization that need to be taken into consideration.*

Reply: We thank the reviewer for the kind advise. The referred study presents the structural evolution of the polycrystalline Cu thin film under CO reduction conditions through *in situ* grazing incidence X-ray diffraction (GIXRD) with high time resolution and surface sensitivity (*ACS Energy Lett.*, 2019, 4, 803–804). They also evidenced the CO-promoted reconstruction to preferential (100) faceting, in agreement with STM studies. In short, this work demonstrated a brilliant and powerful technique for providing dynamic compositional and structural information of poly-Cu using *in situ* GIXRD measurement. Regretfully, the *in situ* GIXRD measurement is currently beyond our ability due to the technology limitation and the scarce synchrotron beam time. We instead conducted the *ex situ* GIXRD measurement, which could demonstrate a dominant exposure of (110) facets with additive after CO₂RR (Fig. 2d). For more insight studies on structural evolution process under catalytic process, *in situ* GIXRD is a powerful tool. We have added related descriptions in the revised manuscript (Line 145, Page 8).

“Grazing incidence X-ray diffraction (GIXRD) patterns show

Fig. 2 | Characterization of the poly-Cu electrodes in different electrolytes. a, SEM image of a poly-Cu electrode after electrocatalysis in the EDTMPA-added electrolyte. **b**, **c**, TEM image (**b**) and HRTEM image (**c**) of an electrodeposited Cu TEM grid after electrocatalysis in the EDTMPA-added electrolyte. Inset in **c** shows the corresponding SAED pattern. **d**, GIXRD patterns of poly-Cu electrodes before/after electrocatalysis in electrolytes with and without EDTMPA. **e**, OH⁻ electroadsorption profiles on the poly-Cu electrodes after electrocatalysis in electrolytes with and without EDTMPA at a sweep rate of 100 mV s⁻¹ in 1 M KOH. **f**, DFT-calculated adsorption energies of an MPA molecule on Cu(110), Cu(100) and Cu(111).

Specific Comment R3-2: *The facile oxidation of copper electrode surfaces presents a challenge when correlating ex-situ structure analysis with in-situ catalytic performance. Can the authors comment on the possible role of surface oxidation and the impact on structure analysis?*

Reply: We thank the reviewer for this reasonable concern. Previous works have shown that the amorphous active sites are highly susceptible to concurrent oxidation and recrystallization upon exposure to ambient conditions (*Proc. Natl. Acad. Sci. USA*, 2020, 117, 9194–9201). Therefore, in our study, the post-electrolysis electrodes were washed with deoxygenated DI water, following by blow-dry with N₂ as soon as possible, and were carefully preserved under Ar atmosphere. Since there are no Cu₂O/CuO signals under our ex situ measurements (XPS, XRD, TEM), it is

reasonable to conclude that our structure analysis is basically unaffected by the surface oxidation. We have added these detailed information to Methods in the revised manuscript (Line 401–405, Page 22).

“In order to avoid oxidation before *ex-situ* characterization, all the Cu electrodes after electrocatalysis were immediately washed with deionized water saturated by N₂, following by blow-dry with N₂ as soon as possible, and finally were carefully preserved in a glove box under Ar atmosphere.”

Specific Comment R3-3: *The SEM images provided in the main text and the supporting information show that the presence of the EDTMPA additive appears to impact the morphology of the surface, but it's difficult to interpret these images in support of the hypothesized addition of more (110) facets with EDTMPA. Both surfaces undergo macroscopic morphology change, but the SEM images are inadequate to unambiguously support this. The features shown in the images are 10s of nm, from which one can't make direct interpretations of the surface construction or enhanced presence of 110 facets.*

Reply: We thank the reviewer for this reasonable concern. It is true that although SEM images could prove the surface reconstruction, they are inadequate to demonstrate a dominant exposure of (110) facets. We further conducted TEM and GIXRD to support the point, which can be seen in Fig. 2b–d and Supplementary Figs. 4–6. We also have added related descriptions in the revised manuscript (Line 132–145, Page 8).

“..... we used electrodeposited Cu TEM grids as poly-Cu electrodes for the CO₂RR and probed their crystal structure before and after electrocatalysis. Representative TEM images of the Cu grid after electrocatalysis in the EDTMPA-added electrolyte show a lot of Cu nanocrystals with hexagonal and cubic outlines, in good agreement with the ideal projections of a rhombic dodecahedral model bounded by {110} facets from different directions²⁷ (Fig. 2b and

Supplementary Fig. 4). The Cu rhombic dodecahedrons are further confirmed by the high-resolution TEM (HRTEM) image and the corresponding selected area electron diffraction (SAED) pattern of the equilateral hexagonal projection shape of Cu nanocrystals along $[111]^{27}$ (Fig. 2c). However, irregular Cu nanoparticles without any preferential surface orientation are merely observed for the electrodeposited Cu TEM grids before and after electrocatalysis in the EDTMPA-free electrolyte (Supplementary Figs. 5, 6).

Grazing incidence X-ray diffraction (GIXRD) patterns show

Fig. 2 | Characterization of the poly-Cu electrodes in different electrolytes. **a**, SEM image of a poly-Cu electrode after electrocatalysis in the EDTMPA-added electrolyte. **b**, **c**, TEM image (**b**) and HRTEM image (**c**) of an electrodeposited Cu TEM grid after electrocatalysis in the EDTMPA-added electrolyte. Inset in **c** shows the corresponding SAED pattern. **d**, GIXRD patterns of poly-Cu electrodes before/after electrocatalysis in electrolytes with and without EDTMPA. **e**, OH^- electroadsorption profiles on the poly-Cu electrodes after electrocatalysis in electrolytes with and without EDTMPA at a sweep rate of 100 mV s^{-1} in 1 M KOH. **f**, DFT-calculated adsorption energies of an MPA molecule on Cu(110), Cu(100) and Cu(111).

Supplementary Fig. 4 | TEM image of an electrodeposited Cu TEM grid after electrocatalysis in the electrolyte with EDTMPA.

Supplementary Fig. 5 | TEM characterization of the as-electrodeposited Cu TEM grid. a, b, TEM images. c, The corresponding SAED pattern of b.

Supplementary Fig. 6 | TEM characterization of an electrodeposited Cu TEM grid after electrocatalysis in the electrolyte without EDTMPA. a, b, TEM images. c, The corresponding SAED pattern of b.

Specific Comment R3-4: *The TEM image of a copper TEM grid doesn't add much to support the electrochemical measurements or the connection between the*

hypothesized surface reconstruction and catalytic activity. As the Cu TEM grids are polycrystalline, one can scan around the sample to find a variety of edge faceting, so picking an image that shows a grain with a (110) surface surrounded by other (hkl) surfaces doesn't add much.

Reply: We thank the reviewer for this reasonable suggestion. As Cu grids are too thick to observe under TEM, we preliminarily electrodeposited a thin layer of polycrystalline Cu dendrites on Cu grids using large current to better identify the difference before and after CO₂RR. The results have been provided in Fig. 2b,c and Supplementary Figs. 4–6. The newly added TEM and SAED images show the most of grains with (110) surface for the post-electrolysis Cu grid with EDTMPA, whereas no preferential exposure of crystal faces is found for the one without EDTMPA. The results give sound evidence of the predominant exposure of (110) facets. For details, please refer to our reply to the comment R3-3.

Specific Comment R3-5: *The XRD patterns in Figure 2d show that the film treated with EDTMPA shows stronger reflection of the (110) planes, with a stronger (110)/(111) ration than the bulk (and presumably the powder reference pattern which should also be included). However, one has to be careful with the XRD analysis, to really be sure that the observed change in reflection ratios is a surface effect, the XRD measurements should be done in grazing incidence configuration (see e.g., Scott et al. ACS Energy Letters 2019, 4 (3), 803-804. DOI: 10.1021/acseenergylett.9b00172)*

Reply: We thank the reviewer for the kind suggestion. Grazing incidence XRD results have been provided in Fig. 2d in the revised manuscript. For details, please refer to our reply to the comment R3-1.

Specific Comment R3-6: *The authors hypothesize a mechanistic explanation that includes three main effects of the additive on the observed catalytic performance: (1) surface reconstruction, (2) mediating proton-feeding, and (3) *CHO stabilization.*

Whereas the paper provides a thorough analysis of the effect of EDTMPA, there is a missed opportunity to enhance the impact and shared insights of this paper by connecting the hypothesized mechanism with the underlying molecular interactions of the EDTMPA and the copper surface.

R3-6-1: *What is it about EDTMPA that enables these unique interactions?*

Reply: We thank the reviewer for this question. Firstly, EDTMPA can be coordinated on Cu surface through Cu-O interactions, which has been verified by *in situ* Raman and DFT calculations (Supplementary Figs. 8, 12). Since the (110) planes with more terminal Cu dangling bonds compared with (100) and (111), EDTMPA has a stronger adsorption energy with (110) than (100) and (111), which is confirmed by DFT calculations (Fig. 2f), leading to the *in-situ* selective growth of (110) facets (Fig. 2a–e). Secondly, EDTMPA is a phosphonic acid, and thus the adsorbed EDTMPA can provide abundant *H to Cu(110) surface, which is indicated by electrochemical tests (Fig. 4a, b) and DFT calculations (Fig. 4c, d). Finally, the adsorbed EDTMPA also stabilizes the *CHO through hydrogen bonds (Fig. 4e and Supplementary Figs. 16, 17), and thus improve the kinetics of CH₄ formation (Fig. 4f).

R3-6-2: *Would this effect also hold for other phosphonic acids or the carbonate analog (EDTA)?*

Reply: We thank the reviewer for this good suggestion. Since MPA is not currently in stock, we selected another small phosphonic acid, methylenediphosphonic acid (MDPA), a two phosphonic counterpart of MPA. We conducted both electrochemical tests and structural characterizations as are shown in the Supplementary Figs. 19, 20 in the revised manuscript. As can be seen in the SEM and XRD results (Supplementary Fig. 19a, c), reconstruction toward (110) facets also took place with the addition of 8 ppm MDPA, indicating the selective surface-capping effect of phosphonic acid molecules on the formation of Cu(110) surface. For the

electrochemical performance, the initial high CH₄ FE (50%) and partial current density (16 mA cm⁻²) suffered from rapid decay accompanied with increased HER (Supplementary Fig. 20a, b), indicating MDPA's promotional effect on H₂ production. The CH₄ activity decline with MDPA additive may be due to the smaller skeleton structure of the MDPA molecule and thus more surface coverage per unit compared with EDTMPA, which could provide excessive *H species for HER and limit CO₂ transportation (insufficient steric hindrance).

Poly-Cu with an 8 ppm EDTA-added electrolyte shows a basically stable CH₄ FE (~50%) and partial current density (18 mA cm⁻²) without rapid decay over 1 h electrolysis, but has a high H₂ FE (~35%) (Supplementary Fig. 20c, d). This could be ascribed to the abundant *H supply by four carboxyl groups in EDTA similar to four phosphate groups in EDTMPA. The phenomenon further discloses the key role of surface *H played in protonation of *CO elementary step for stable methane production. However, due to the lack of reconstruction, as is indicated by SEM and XRD results (Supplementary Fig. 19b, c), the electrochemical performance shows a relatively lower CH₄ activity compared with EDTMPA-added case. This highlights another unique role of EDTMPA additive in improving *CO coverage on the reconstructed (110) facets and thus improves methane formation. We have added the figures as Supplementary Figs. 19, 20 to the revised Supplementary Information and described in the revised manuscript (Line 349–369, Page 20 and 21).

“To illustrate the effects of additives on the surface reconstruction and electrocatalytic performance, we also investigated two analogues of EDTMPA, namely, methylenediphosphonic acid (MDPA) and ethylenediamine tetraacetic acid (EDTA), as electrolyte additives for the CO₂RR. According to SEM and X-ray diffraction (XRD) results, MDPA has an effect on the generation of Cu(110) surface like EDTMPA (Supplementary Fig. 19a, c). However, the CH₄ FE suffers from rapid decay accompanied with increased H₂ FE (Supplementary Fig. 20a), and their partial current densities are both decreased during one-hour electrocatalysis in the MDPA-added electrolyte (Supplementary Fig. 20b). This may be largely attributed to

the smaller space structure of MDPA molecules and the resulting higher coverage on Cu surface compared with EDTMPA, which results in an excessive proton supply and limited CO₂ transportation. A substantial stability of the CO₂-to-CH₄ conversion is achieved after adding EDTA into the electrolyte, although the CH₄ FE (~50%) and partial current density (18 mA cm⁻²) are lower than those in the EDTMPA-added case (Supplementary Figs. 1 and 20c, d). This can be attributed to the similar proton-feeding capability of the carboxyl groups in EDTA with the phosphate groups in EDTMPA. The inferior performance for CH₄ production is due to the inability of EDTA to induce the atomic re-arrangement of Cu surface to generate Cu(110) like EDTMPA and MDPA (Supplementary Fig. 19b, c). Therefore, it can be concluded that the additives should be rational selected according to the desired products, although the selecting principles are still under exploration.”

Supplementary Fig. 19 | Characterization of the poly-Cu electrodes with different additives. a, b, SEM images of poly-Cu electrodes after electrocatalysis in the electrolyte with MDPA (a) and EDTA (b). c, XRD patterns of poly-Cu electrodes after electrocatalysis in the electrolytes with MDPA and EDTA.

Supplementary Fig. 20 | CO₂ electroreduction performance with different additives. a, b, FEs (a) and partial current densities (b) of various products at -1.0 V versus RHE in the electrolytes with 8 ppm MDPA. c, d, FEs (c) and partial current densities (d) of various products at -1.0 V versus RHE in the electrolytes with 8 ppm EDTA.

R3-6-3: *The fact that this effect is pronounced at such low concentrations (8ppm) raises questions about the role of other additives (intentional) or contaminations (unintentional and unknown). Can the authors comment on the possible role / presence of other surface active species in the electrolyte?*

Reply: We thank the reviewer for the suggestion. All chemicals selected in our work are ultra-pure, and we also purified our electrolytes via pre-electrolysis process before testes to further remove any contaminations. Moreover, we explored control experiments using the electrolyte without additives to confirm our experimental results and conclusions are reliable.

Specific Comment R3-7: *Minor point: The abstract states " Electrocatalytic CO2 reduction to value-added hydrocarbon products using metallic 17 copper (Cu) catalysts is a *sustainable approach to facilitate carbon neutrality. "; the reviewer suggests revising this as 'a potentially sustainable' approach as there still persist several hurdles to translate scientific discoveries to a truly sustainable, and deployable (scalable, economically viable etc.) technology*

Reply: We thank the reviewer for this good suggestion. We have made corresponding modifications in the revised manuscript (Line 20, Page 1).

“..... is a potentially sustainable approach ”

REVIEWERS' COMMENTS

Reviewer #1 (Remarks to the Author):

The authors have adequately revised the manuscript based on the reviewer comments, so I am happy to support publication of this work.

Reviewer #2 (Remarks to the Author):

The authors have addressed all my concerns. I believe the manuscript is now ready for nature communications.

Reviewer #3 (Remarks to the Author):

The authors have done a good job in systematically addressing the concerns raised in the initial review. One point I would recommend to revise: Since the electrolyte effects are noticeable at low concentrations (8ppm) it would be helpful to include a note to the general community along the lines of "The fact that the electrolyte effects are noticeable at low concentrations highlights that researchers need to pay careful attention to the presence of additives (intentional) or contaminations (unintentional or unknown) in their experiments."

Response to the reviewers' comments

Reviewer 1:

The authors have adequately revised the manuscript based on the reviewer comments, so I am happy to support publication of this work.

Reply: We appreciate the reviewer for his/her positive comments and recommendation for publication in *Nature Communications*.

Reviewer 2:

The authors have addressed all my concerns. I believe the manuscript is now ready for nature communications.

Reply: We appreciate the reviewer for his/her positive comments and recommendation for publication in *Nature Communications*.

Reviewer 3:

The authors have done a good job in systematically addressing the concerns raised in the initial review. One point I would recommend to revise: Since the electrolyte effects are noticeable at low concentrations (8ppm) it would be helpful to include a note to the general community along the lines of "The fact that the electrolyte effects are noticeable at low concentrations highlights that researchers need to pay careful attention to the presence of additives (intentional) or contaminations (unintentional or unknown) in their experiments."

Reply: We appreciate the reviewer for his/her positive comments and recommendation for publication in *Nature Communications*. Following the reviewer's

nice suggestion, we have added a note to the Discussion section in the revised manuscript. Please see **Line 15–18, Page 16**: “Therefore, it can be concluded that the additives should be rational selected according to the desired products, although the selecting principles are still under exploration. Incidentally, the fact that the electrolyte effects are noticeable at low concentrations highlights that researchers need to pay careful attention to the presence of additives (intentional) or contaminations (unintentional or unknown) in their experiments.”